# MDSGEN: FAST AND EFFICIENT MASKED DIFFUSION TEMPORAL-AWARE TRANSFORMERS FOR OPEN-DOMAIN SOUND GENERATION

**Trung X. Pham**[*], **Tri Ton**[*]**, Chang D. Yoo**[†]
Korea Advanced Institute of Science and Technology (KAIST)
{trungpx, tritth, cd_yoo}@kaist.ac.kr

## ABSTRACT

We introduce `MDSGen`, a novel framework for vision-guided open-domain sound generation optimized for model parameter size, memory consumption, and inference speed. This framework incorporates two key innovations: (1) a redundant video feature removal module that filters out unnecessary visual information, and (2) a temporal-aware masking strategy that leverages temporal context for enhanced audio generation accuracy. In contrast to existing resource-heavy Unet-based models, `MDSGen` employs denoising masked diffusion transformers, facilitating efficient generation without reliance on pre-trained diffusion models. Evaluated on the benchmark VGGSound dataset, our smallest model (5M parameters) achieves 97.9% alignment accuracy, using $172\times$ fewer parameters, 371% less memory, and offering $36\times$ faster inference than the current 860M-parameter state-of-the-art model (93.9% accuracy). The larger model (131M parameters) reaches nearly 99% accuracy while requiring $6.5\times$ fewer parameters. These results highlight the scalability and effectiveness of our approach. The code is available at https://bit.ly/mdsgen.

## 1 INTRODUCTION

Vision-guided audio generation has gained significant attention due to its crucial role in Foley sound synthesis for the video and film production industry (Ament, 2014). This paper focuses on Video-to-Audio (V2A) generation, a key task not only for adding realistic sound to silent videos created by emerging text-to-video models (Blattmann et al., 2023; Khachatryan et al., 2023; Huang et al., 2024; Ouyang et al., 2024) but also for enhancing practical applications in professional video production. Sound generation is essential for creating immersive experiences and achieving seamless audio-visual synchronization. However, achieving both semantic alignment and temporal synchronization in V2A remains a significant challenge. Previous approaches, such as GAN-based methods (Chen et al., 2020b) and Transformer-based autoregressive models (Iashin & Rahtu, 2021),

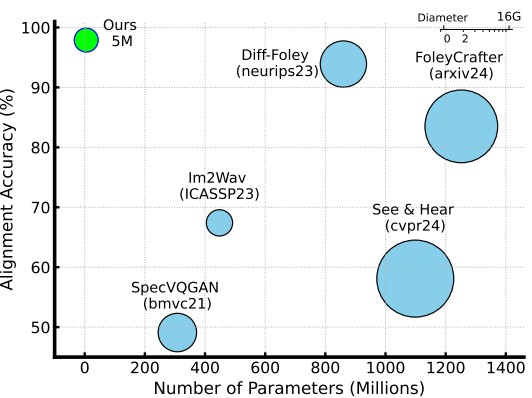

Figure 1: **Alignment Score.** Comparison with SOTA audio generation methods on the VGGSound test set. The diameter of each circle represents the memory usage during inference.

have struggled with synchronizing audio to content while maintaining relevance. Diff-Foley (Luo et al., 2023) improved this by employing contrastive learning for video-audio alignment and leveraging diffusion models, achieving impressive sound quality. Other methods like See and Hear (Xing et al., 2024) and FoleyCrater (Zhang et al., 2024) utilize large pre-trained models for high-quality

---

[*]Equal Contribution
[†]Corresponding Author

audio generation. However, these models rely on hundreds of millions of parameters. In contrast, our work demonstrates that a much smaller model can deliver high performance (see Fig. 1). Most existing approaches rely on Unet architectures, which present scalability limitations. Additionally, current methods often use video features that include redundant information.

In contrast, we propose `MDSGen`, a novel framework for open-domain sound synthesis based on a pure Transformer architecture. `MDSGen` incorporates a temporal-aware masking scheme and a redundant feature removal module, enabling it to achieve superior performance while being significantly more efficient. Further analysis highlights the effectiveness of our approach with compelling evidence of its advantages. Our contributions are as follows:

- We introduce a simple, lightweight, and efficient framework for open-domain sound generation using masked diffusion transformers, delivering high performance.
- Our approach implements Temporal-Awareness Masking (TAM), specifically designed for audio modality, in contrast to spatial-aware masking of the existing work, leading to more effective learning.
- We identify inefficiencies in the existing approach that fail to remove redundant video features. Our Reducer module learns to selectively resolve these redundancies, producing more refined features for improved audio generation.
- We validate our method on the benchmark datasets VGGSound and Flickr-SoundNet, surpassing state-of-the-art approaches across multiple metrics, with particularly significant improvements in alignment accuracy and efficiency, specifically in model parameters, memory consumption, and inference speed.

## 2 RELATED WORKS

### 2.1 OPEN-DOMAIN SOUND GENERATION

**Auto-regressive Transformer-based Approach.** Key works in this area include SpecVQGAN (Iashin & Rahtu, 2021), which uses a cross-modal Transformer to generate sounds from video tokens auto-regressively, and Im2Wav (Sheffer & Adi, 2023), which conditions audio token generation on CLIP features. However, these methods suffer from slow inference speeds due to their sequential generation process and limited vision-audio alignment, negatively impacting performance.

**Diffusion-based Approach.** To overcome these limitations, Diff-Foley (Luo et al., 2023) introduced a two-stage method that enhances semantic and temporal alignment via contrastive pre-training on aligned video-audio pairs, followed by latent diffusion for improved inference efficiency. Similarly, See and Hear (Xing et al., 2024) utilizes ImageBind (Girdhar et al., 2023) and AudioLDM (Liu et al., 2023) for various audio tasks, while FoleyCrafter (Zhang et al., 2024) combines a pre-trained text-to-audio model with a ControlNet-style module (Zhang et al., 2023) for high-quality, synchronized Foley generation. Although these diffusion approaches show promise, they often rely on large models with hundreds of millions of parameters and predominantly utilize U-Net architectures, leaving the potential of transformer-based architectures largely untapped. Our proposed method leverages diffusion transformers (Peebles & Xie, 2023) and masking techniques for efficient learning. It also addresses the issue of redundant video features in Diff-Foley (Luo et al., 2023), which hinders further improvements in audio generation.

### 2.2 LATENT MASKED DIFFUSION TRANSFORMERS

The Denoising Diffusion Transformer (DiT) introduced by Peebles & Xie (2023) replaces the traditional U-Net with a fully transformer-based architecture for latent diffusion, demonstrating remarkable performance in large-scale image generation on ImageNet. Following this, Gao et al. (2023) proposed the Masked Diffusion Transformer (MDT), which enhances ImageNet generation through spatial context-aware masking. Inspired by MDT, Pham et al. (2024) developed X-MDPT, using cross-view masking to establish correspondence between pose and reference images for improved person image generation. Additionally, MDT-A2G (Mao et al., 2024) explored masked diffusion transformers for gesture generation, while QA-MDT (Li et al., 2024) adapted this technique for music generation. In contrast to these works, we focus on lightweight masked diffusion models

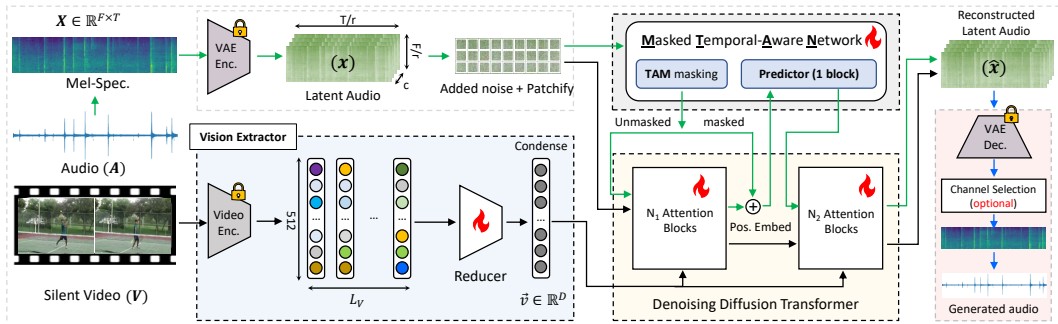

Figure 2: **Overview of the proposed highly-efficient `MDSGen` framework**, utilizing denoising masked diffusion transformers to efficiently learn video-conditional distributions for audio generation, replacing traditional Unet-based methods. The fire icon denotes trainable modules, and the locked icon denotes frozen ones. Green arrows → denote branches used only during training, blue arrows → are for only inference, and black arrows → are used in both training and inference.

for video-guided audio generation, introducing temporal-aware masking for audio and a design that removes redundant video features to enhance generation effectiveness.

## 2.3 MASKED DATA IN THE AUDIO MODALITY

Several works have explored masking techniques for audio processing. MaskVAT (Pascual et al., 2024) introduces a V2A system that integrates a full-band audio codec, masked generative modeling, and multi-modal features to enhance audio quality, semantic alignment, and temporal synchronization. SoundStorm (Borsos et al., 2023) employs the non-autoregressive MaskGIT (Chang et al., 2022) approach for efficient text-to-audio generation. Similarly, VamNet (Garcia et al., 2023) applies MaskGIT to music generation, while (Bai et al., 2022) uses masking in the pixel space of mel-spectrograms for non-diffusion-based text-to-audio tasks. These approaches differ fundamentally from our diffusion-based framework, which applies masking in the latent space with added Gaussian noise. AudioMAE (Huang et al., 2022) and SpecAugment (Park et al., 2019) share similarities with our method in employing masking for audio data. However, key distinctions exist: both focus on masking in the pixel space of clean mel-spectrograms for representation learning in downstream recognition tasks. In contrast, our approach utilizes masking in the latent space of a VAE within the diffusion framework, targeting audio generation.

## 3 METHOD

We aim to develop a simple yet effective framework for vision-guided sound generation using transformers, addressing the limitations of existing approaches that rely on traditional U-Net architectures, which are less scalable and efficient. Our framework, illustrated in Fig. 2, consists of a novel **Vision Extractor** with a learnable **Reducer** that captures essential information from video input to generate a concise conditional output for the denoising diffusion process. Next, a **Denoising Diffusion Transformer** maps Gaussian noise to sound distributions using extracted visual features. We also introduce a **Masked Temporal-Aware Network (MTANet)** for regularization, boosting performance. Finally, channel selection for mel-spectrograms, which enhances results with image VAEs, is optional.

### 3.1 DENOISING DIFFUSION TRANSFORMER

Our method supports both audio- and image-based VAEs. For instance, we describe using an image VAE (Luo et al., 2023; Chen et al., 2024), and for audio VAEs like AudioLDM, we adjust the three channels to one. We adopt the DiT backbone introduced by Peebles & Xie (2023) for denoising diffusion training. Given an audio signal $\mathbf{A} \in \mathbb{R}^{L_\mathbf{A}}$ of length $L_\mathbf{A}$ and a silent video $\mathbf{V} \in \mathbb{R}^{L_V \times 3 \times 224 \times 224}$ of length $L_V$, the audio is first transformed into a mel-spectrogram $\mathbf{X} \in \mathbb{R}^{128 \times 512}$, while the video is encoded into $\mathbf{v} \in \mathbb{R}^{L_V \times 512}$ and further reduced to $\vec{v} \in \mathbb{R}^{1 \times D}$. The mel-spectrogram is repeated across 3 channels, forming $\mathbf{X}' \in \mathbb{R}^{3 \times 128 \times 512}$, and passed through the

VAE from Stable Diffusion (Rombach et al., 2022; Luo et al., 2023) to obtain a latent embedding $\mathbf{x} \in \mathbb{R}^{4 \times 16 \times 64}$. This latent representation is patched and tokenized into image tokens using a patch size of $p = 2$ (DiT's default), resulting in $\mathbf{x}' \in \mathbb{R}^{256 \times D}$, where $L_{\mathbf{x}'} = 256$ and $D = 768$ for the B-size model. These tokens are then fed into the $N = N_1 + N_2$ self-attention layers of the Transformer to predict the noise $\epsilon$ added to the latent $\mathbf{x}$. Conditioned on the video encoding $\vec{v}$, the Transformer model $\phi$ learns the distribution $p_\phi(\mathbf{x}|\vec{v})$. During training, Gaussian noise $\epsilon \in \mathcal{N}(0, \mathbf{I})$ is added to the latent $\mathbf{x}$ to generate $\mathbf{x}_t$ at timestep $t \in [1, T]$. The overall training objective is:

$$\mathcal{L}_{\sum} = \mathbb{E}_{\mathbf{x}, \vec{v}, \epsilon} \|\epsilon - \epsilon_\phi(\mathbf{x_t}, \vec{v}, t)\|^2 + \lambda \mathbb{E}_{\mathbf{x}, \vec{v}, \epsilon} \|\epsilon - \epsilon_\phi(\mathcal{M}_\phi(\mathbf{x_t}), \vec{v}, t)\|^2. \tag{1}$$

Here, $\lambda$ is the balance factor between the standard denoising diffusion loss (the first term in Eq. 1) and the masking loss (the second term), with $\lambda = 1.0$ for optimal performance. The masking function $\mathcal{M}_\phi$, which includes the MTANet introduced later, applies temporal-aware masking. During inference, given a silent video, the model starts from Gaussian noise (no audio provided), and the predicted latent $\hat{\mathbf{x}} \in \mathbb{R}^{4 \times 16 \times 64}$ is iteratively denoised and decoded by the VAE decoder to recover the mel-spectrogram $\widehat{\mathbf{X}}_{RGB} \in \mathbb{R}^{3 \times 128 \times 512}$. Channel selection refines this into $\widehat{\mathbf{X}} \in \mathbb{R}^{128 \times 512}$, and the final waveform is reconstructed from the mel-spectrogram using the Griffin-Lim (Griffin & Lim, 1984), or neural Hifi-GAN vocoder Kong et al. (2020).

## 3.2 VISION EXTRACTOR

The second key component of our framework is the Vision Extractor with a learnable Reducer network that aligns video features with audio while condensing temporal information. We leverage the pre-trained CAVP model from (Luo et al., 2023), which was trained on Audioset using contrastive loss to extract video features aligned with audio. However, we identified that the CAVP features contain redundancies that could negatively impact generation quality. Diff-Foley (Luo et al., 2023) linearly maps original feature dimensions from $\mathbf{v} \in \mathbb{R}^{L_V \times 512}$ to $\mathbf{v} \in \mathbb{R}^{L_V \times 768}$ and retains this full dimensionality during the diffusion process via cross-attention, with $L_V$ is the video feature length. Our approach reduces the dimensionality to $\vec{v} \in \mathbb{R}^{1 \times 768}$, offering more concise and efficient information for denoising diffusion. Specifically, we project the encoded features $L_V \times 512$ through a multi-layer perceptron (MLP) into the transformer feature space ($L_V \times 768$ for the size B-model). These features are then passed through a reducer module, an $1 \times 1$ convolutional layer, which condenses the high-dimensional features into a lightweight form $\vec{v} \in \mathbb{R}^{1 \times 768}$. This compact representation is integrated into the denoising diffusion process through Adaptive LayerNorm (AdaLN) modulation. Our method minimizes redundant features that could lead to overfitting, as shown in the train/test alignment accuracy gap Appendix Sec. A.2. Our analysis shows that the $L_V$-**frame input features share over** $90\%$ **similarity**, indicating considerable redundancies.

**Intuition.** Our simple yet effective reducer design treats the temporal dimension of video ($L_V = 32$) as feature channels and performs a non-linear projection to a single channel, functioning similarly to channel attention by weighting important channels and summing them. This acts as a bottleneck that distills and distributes video temporal information across the 768 dimensions, aligning better with each audio token, which also has a 768-dimensional space. This approach, combined with the transformer network, significantly improves alignment accuracy up to approximately 99%.

## 3.3 AUDIO MASKED TEMPORAL-AWARE NETWORK

Thirdly, we introduce a novel technique that exploits the sound information's natural characteristic: the temporal sense. The existing masking, **Spatial-Aware Mask** (SAM) proposed by MDT (Gao et al., 2023) is designed for image data to learn the spatial context within the image. But here, in the audio data (represented by mel-spectrogram with 2D data), the SAM masking method yields a sub-optimal solution because it cannot model the exact nature of temporal meaning in the audio data. To overcome this limitation, we propose the **Temporal-Aware Mask** (TAM) strategy instead of SAM, which tries to mask the whole set of tokens along the temporal

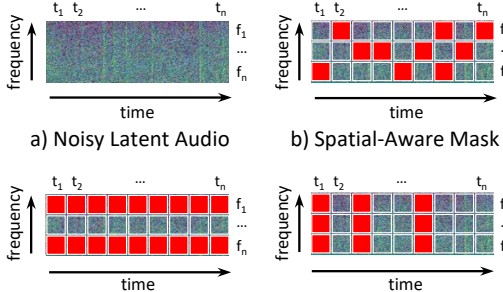

a) Noisy Latent Audio    b) Spatial-Aware Mask

c) Frequency-Aware Mask    d) Temporal-Aware Mask

Figure 3: **Audio Masking Strategies**. Here, the red square red-square is the learnable mask token.

dimension. As shown in the ablation section, this novel masking helps significantly boost performance in all metrics compared to the existing method SAM specified for image data as shown in Fig. 3. Interestingly, despite this simple strategy, it can help the denoising transformer models learn to generate audio much better than random masks as used in existing MDT designed for image data.

During training, we mask with $\eta_{\mathbf{m}}\%$ of temporal tokens, we feed only the visible tokens to the $N_1$ blocks of the transformer: $\mathbf{o}_1 = \text{Encoder}_{N_1}(\mathbf{x}' \odot (1 - \mathbf{m}))$ ($\mathbf{m}$ is a mask matrix and $\odot$ denotes element-wise multiplication) and in the MTANet we concatenate the resulting tokens with the learnable mask tokens $\mathbf{M}$ and feed into a single block of predictor (referred to as side-interpolator in MDT) to achieve the full latent tokens before feeding into the final $N_2$ self-attention blocks of the transformer: $\mathbf{o}_2 = \text{Decoder}_{N_2}(\text{cat}(\mathbf{o}_1, \mathcal{M}_\phi(\mathbf{M} * \mathbf{x}' \odot (\mathbf{m}))))$. Here we find that different from the design for ImageNet with $N_2 = 2$ in the default MDT, we use $N_2 = 4$ which gives a better performance for audio data. After training, the masked modeling branch is discarded, maintaining only its positional embedding for inference.

### 3.4 CLASSIFIER-FREE GUIDANCE

We adopt the dynamic Classifier-Free Guidance (CFG) method from previous works on masked diffusion transformer models (Gao et al., 2023) used in ImageNet. However, unlike image tasks, we find that in audio generation, the optimal CFG value is between 5 and 6, with a power scale of 0.01. Notably, Classifier-Guidance (CG) has been shown to significantly boost performance in Diff-Foley (Luo et al., 2023), where their method relies heavily on CG for optimal results. In contrast, our approach without CG surpasses Diff-Foley (CFG+CG) across multiple metrics. While incorporating CG improves our framework in terms of alignment accuracy and KL, it does not enhance other metrics. Hence, for simplicity, we omit CG in most of our experiments.

### 3.5 VAE CHOICE FOR MEL-SPECTROGRAM

Our method supports both Audio- and Image-trained VAEs. Ablation finds that audio quality varies across the RGB output channels of image-trained VAEs. Since the mel-spectrogram is 2D, we duplicate it into three channels for Stable Diffusion VAE. At the decoding stage, the VAE outputs three channels: $\widehat{\mathbf{X}}_{RGB} \in \mathbb{R}^{3 \times 128 \times 512}$, with $\widehat{\mathbf{X}}_{RGB}[i, :, :] \in \mathbb{R}^{128 \times 512}, i \in \{0, 1, 2\}$ representing the R, G, and B channels. Diff-Foley (Luo et al., 2023) used the R channel as output, but our empirical tests consistently show that the G channel performs better. However, when using the audio VAE, *i.e.* AudioLDM VAE, channel selection is no longer required.

## 4 EXPERIMENTS

### 4.1 DATASET AND EVALUATION METRICS

**(i) Dataset.** We evaluate our method on the VGGSound dataset (Chen et al., 2020a), using the original train/test splits with 175k and 15k samples, respectively, and on the Flick-SoundNet dataset Aytar et al. (2016) with 5k test samples. **(ii) Metrics.** We first use the same metrics as prior work (Luo et al., 2023), including FID, IS, KL, and Alignment Accuracy, using their provided scripts for Align. Acc. and SpecVQGAN code for FID, IS, and KL. Second, we assess general vision-audio alignment in the Image2Audio task using CIoU and AUC metrics with scripts from (Mo & Morgado, 2022). Third, we compare efficiency using parameter count, memory usage, and inference speed. Lastly, we provide the FAD scores and MOS results from human evaluations in the Appendix A.

### 4.2 IMPLEMENTATION DETAILS

All models are trained and tested on a single A100 GPU (80GB) with a batch size of 64 and a learning rate of 5e-4, using the Adan optimizer (Xie et al., 2024) for faster training. Unlike MDTv2 (Gao et al., 2023), we skip the macro-style of side interpolator design, as it was ineffective for our task, and instead use a simple self-attention block at decoder layer 4. Video-audio pairs are truncated to 8.2 seconds before encoding, following (Luo et al., 2023). Our model comes in three main variants: Tiny (5M), Small (33M), and Base (131M), with the Large (460M) variant showing overfitting. We primarily focus on the T, S, and B models.

### 4.3 MAIN RESULTS

**A. VGGSound dataset.** Compared to state-of-the-art approaches, our method significantly outperforms all competitors' alignment accuracy while being far more efficient regarding parameters and inference speed (Tab. 1). Alignment accuracy, a metric introduced by (Luo et al., 2023), assesses synchronization and audio-visual relevance using a separate classifier trained to predict real audio-visual pairs. Remarkably, our Transformer-based model, MDSGen-Tiny (5M), trained from scratch, achieves 97.9% accuracy, surpassing the second-best Diff-Foley (860M), which is 172× larger and depends on a backbone of Stable Diffusion pre-trained on billion image-text pairs.

As shown in Tab. 1, Diff-Foley struggles without a pre-trained backbone, with a significant drop to FID 16.98 and IS 24.91. In contrast, our smallest model, MDSGen-T (5M), trained from scratch, achieves FID 14.18 and IS 37.51, emphasizing the overfitting issues of heavy U-Net-based models compared to our lightweight Transformers. Our larger model, MDSGen-B (131M), achieves state-of-the-art alignment accuracy ($\approx$ 99%) and an IS of 57.12 at 800k steps, though longer training leads to overfitting and declines in other metrics.

Table 1: **Benchmark on VGGSound test.** Generation quality comparison of different approaches. † gets from (Luo et al., 2023), ‡ denotes without pre-trained SDv1.4. * denotes results with pre-trained SDv1.4, we reproduce it using the public checkpoint. "()" in FAD denotes AudioVAE (ImageVAE).

| Method | FAD↓ | FID↓ | IS↑ | KL↓ | Align. Acc.↑ | Time↓ (s) | #Params↓ | Cost↓ |
|---|---|---|---|---|---|---|---|---|
| SpecVQGAN (Iashin & Rahtu, 2021)† | - | **9.70** | 30.80 | 7.03 | 49.19 | 5.47 | 308M | 61× |
| Im2Wav (Sheffer & Adi, 2023) † | - | 11.44 | 39.30 | **5.20** | 67.40 | 6.41 | 448M | 90× |
| Diff-Foley (Luo et al., 2023) †‡ | - | 16.98 | 24.91 | 6.05 | 92.61 | 0.38 | 860M | 172× |
| Diff-Foley (Luo et al., 2023) * | 4.71 | 10.55 | 56.67 | 6.49 | 93.92 | 0.36 | 860M | 172× |
| See and Hear (Xing et al., 2024) | 5.55 | 21.35 | 19.23 | 6.94 | 58.14 | 18.25 | 1099M | 220× |
| FoleyCrafter (Zhang et al., 2024) | 2.45 | 12.07 | 42.06 | 5.67 | 83.54 | 2.96 | 1252M | 250× |
| **MDSGen-T (Ours) 500k** | **2.21 (3.69)** | 14.18 | 37.51 | 6.25 | **97.91** | **0.01** | **5M** | 1.0× |
| **MDSGen-S (Ours) 500k** | **1.66 (2.75)** | 12.92 | 44.38 | 6.29 | **98.32** | **0.02** | 33M | 6.6× |
| **MDSGen-B (Ours) 500k** | **1.34 (2.16)** | 11.19 | 52.77 | 6.27 | **98.55** | **0.05** | 131M | 26.2× |
| **MDSGen-B (Ours) 800k** | - | 12.29 | **57.12** | 6.43 | 91.62 | **0.05** | 131M | 26.2× |

**B. Flickr_SoundNet dataset.** We use the models trained on VGGSound to test on SoundNet dataset to evaluate its generalization. First, through quantitative metrics in the sound source localization task with Flickr-SoundNet (Aytar et al., 2016) test set. Second, qualitatively compare the generated audio across different methods. As shown in Tab. 2, our method outperforms other methods on the CIoU metric (82.01%) closer to the ground truth (83.94%), while the AUC remains comparable (around 55.5%). It shows that our generated audio provided better-aligned features with the visual information to localize the sound source. Diff-Foley performs worst, indicating that it is more overfitting. We provide their visualizations in the Appendix A.

Table 2: **Benchmark on Flick-SoundNet test.** Comparison of different approaches. '**Bold**' and "underline" denote the best and second-best, respectively.

| Method | CIoU↑ | AUC↑ |
|---|---|---|
| Diff-Foley (Luo et al., 2023) | 81.02 | 55.19 |
| See and Hear (Xing et al., 2024) | 81.20 | 55.45 |
| FoleyCrafter (Zhang et al., 2024) | 81.78 | **55.57** |
| **MDSGen-B (Ours) 500k** | **82.01** | 55.51 |
| Ground Truth | 83.94 | 63.60 |

## 5 ABLATION STUDY

We attribute the strong performance of our models to three key factors. First, the **Transformer backbone** enables more effective learning of the audio modality compared to existing Unet-based diffusion methods (Diff-Foley, See and Hear, FoleyCrafter). Second, our innovative **Reducer** module mitigates potential redundancies in the video input. Third, the **temporal masking model** acts as a robust regularizer, further enhancing the Transformer's performance. A detailed analysis of these components is provided in the following sections.

### 5.1 ALIGNMENT ACCURACY: A CONFIDENCE SCORE PERSPECTIVE

We assess the enhancement of alignment accuracy in our method by analyzing confidence scores from the VGGSound test set, using the output of the sigmoid function from the trained classifier

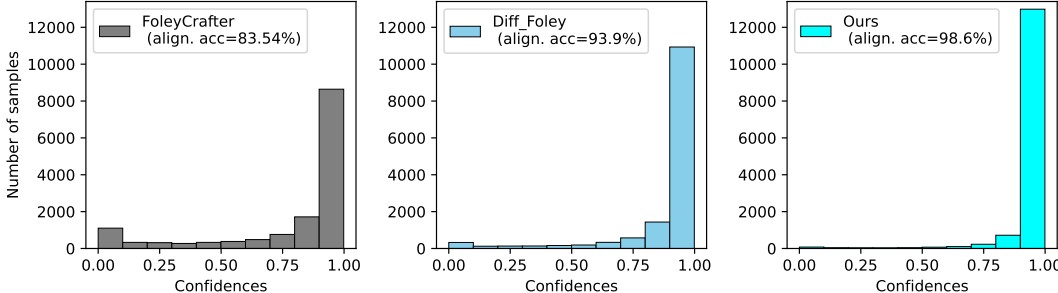

Figure 4: **Confidence Scores.** Compared to FoleyCrafter (left) and Diff-Foley (middle), our method (right) produces many more audio samples with higher confidence that align with their corresponding videos on the VGGSound test set ($\sim$ 15k samples).

that predicts audio-video alignment. As illustrated in Fig. 4, FoleyCrafter (acc=83.54%) produces many low-confidence samples, indicating misalignment. In contrast, Diff-Foley (93.9%) achieves a higher proportion of high-confidence scores. Remarkably, our method reaches an accuracy of 98.6%, significantly increasing the number of high-confidence samples and demonstrating superior audio-video alignment compared to other approaches.

## 5.2 FEATURE DIMENSIONALITY REDUCTION AND REDUNDANT FEATURES IN CAVP

Unlike Diff-Foley, which uses a U-Net-based Stable Diffusion model and incorporates all 32 video frame features for cross-attention, we found that reducing video features from 32 channels to a single channel significantly improves audio generation performance across all metrics

Table 3: **Dimension Reduction.** Compare the original CAVP and Ours's features.

| Video Feat. | Cond. Dim. | FID | IS | KL | Align. Acc. |
|---|---|---|---|---|---|
| Original CAVP | $32 \times 768$ | 13.55 | 50.12 | 6.38 | 96.18 |
| **Reduced (Ours)** | $1 \times 768$ | **11.19** | **52.77** | **6.27** | **98.55** |

(Tab. 3). Diff-Foley's CAVP encodes video features at $32 \times 512$, aligning with the audio's 32-channel representation (also $32 \times 512$) for latent-level alignment via contrastive learning. However, in the second stage, a mismatch arises as the VAE reduces audio dimensions to $4 \times 16 \times 64$ while video expands to $32 \times 768$, leading to redundancy and inefficiencies. Reducing video dimensionality to $1 \times 768$ representation acts as a bottleneck, simplifying learning and enhancing alignment, as supported by our results in the Appendix A.2.

Analysis of the CAVP features in Diff-Foley shows significant redundancy, with cosine similarity averaging **0.9087** for real videos and **0.9233** for identical frames (Tab. 4). These high similarity scores suggest that the feature vectors from multiple frames largely overlap, limiting the model's ability to learn distinctive characteristics essential for effective audio synthesis. This issue worsens when Diff-Foley expands features to a $32 \times 768$ dimension, diluting key traits

Table 4: **Redundant Features.** Cosine similarity between the frame's features.

| Input | Similarity (CAVP) |
|---|---|
| Video | 0.9087 |
| Image | 0.9233 |

for latent diffusion modeling. In contrast, our approach employs a Reducer module to consolidate the 32 video frame features into a 768-dimensional representation, effectively reducing redundancy and enhancing focus on salient features, which improves alignment accuracy and overall performance in audio generation tasks.

## 5.3 DOES NEURAL VOCODER HELPS?

We conducted experiments by training a separate neural vocoder to convert mel-spectrograms back to waveforms. Using the publicly available HiFi-GAN code from GitHub, we trained the vocoder from scratch on the VGGSound dataset, achieving good convergence within three days. As shown in Tab. 5, our method achieves an FAD score of **4.3788** with the simple Griffin-Lim algorithm, outperforming both See-and-Hear (5.5547) and Diff-Foley (6.0810). When enhanced with the HiFi-GAN neural vocoder, our method further improves the FAD score to **2.1610**, achieving state-of-the-art performance on this metric. This result not only highlights the superiority of our approach but also demonstrates its robustness when evaluated beyond the metrics reported in Tab. 1.

Table 5: **FAD on VGGSound test set.** Our method MDSGen-B using the simple Griffin-Lim outperforms the two methods and is state-of-the-art when equipped with a vocoder HifiGAN.

| Method | See-and-hear (neural vocoder) | FoleyCrafter (neural vocoder) | Diff-Foley (Griffin-Lim) | Diff-Foley (neural vocoder) | MDSGen-B (Griffin-Lim) | MDSGen-B (neural vocoder) |
|--------|--------|--------|--------|--------|--------|--------|
| **FAD**↓ | 5.5547 | 2.4554 | 6.0810 | 4.7168 | 4.3788 | **2.1610** |

## 5.4 Subjective Human Evaluation Tests

We conducted a human evaluation by generating 50 audio samples based on 50 videos for each method. Five participants were asked to evaluate each method. Participants were instructed to watch the videos and listen to the corresponding audio, rating each on a scale from 1 to 5 based on the following criteria: 1) Audio Quality (AQ): How good is the sound quality? and 2) Audio-Video Content Alignment (AV): How well does the sound match the video content? The mean opinion scores (MOS) for each metric (ranging from 1 to 5) are presented in Tab. 6. The results from our human evaluation demonstrate that the waveforms generated by our model outperform those of competing methods, receiving higher scores across the evaluation criteria. Participants consistently rated the audio quality and alignment with the video content more favorably for our generated waveforms, indicating superior perceptual performance.

Table 6: **Human Evaluation (MOS).** Our method MDSGen-B equipped with a vocoder HifiGAN. AQ: Audio Quality, AV: Audio-visual content relevance.

| Method | See-and-hear | FoleyCrafter | Diff-Foley | **MDSGen-B (Ours)** | **Ground Truth** |
|--------|--------|--------|--------|--------|--------|
| MOS-AQ↑ | 2.68±0.25 | 3.21±0.23 | 3.29±0.24 | **3.66±0.23** | **4.74 ±0.12** |
| MOS-AV↑ | 2.95±0.20 | 3.44±0.26 | 3.56±0.23 | **3.76±0.21** | **4.62±0.23** |

## 5.5 Masking Diffusion Strategies

We explore various audio-masking methods in diffusion transformers (Fig. 3) and compare them to traditional image-based techniques like SAM used with ImageNet (Gao et al., 2023). Our findings reveal that audio data behaves differently from images (SAM), with incorporating temporal awareness (TAM) into the masking task significantly boosting performance across all metrics (Tab. 7), including a 4-point IS score increase from 48.66 to 52.77. Using DiT without masking leads to suboptimal results across all metrics, highlighting the critical role of masking in model learning. TAM outperforms FAM, as masking in the temporal dimension typically yields better performance

Table 7: **Masking Strategy for Audio Generation.** Comparison of different ways to train diffusion transformer-based models. Masking on temporal audio gives the best performance.

| Masking Method | Mask | Temporal | FID↓ | IS↑ | KL↓ | Align. Acc.↑ (%) |
|--------|--------|--------|--------|--------|--------|--------|
| DiT (Peebles & Xie, 2023) | × | × | 14.55 | 46.11 | 6.51 | 97.12 |
| Random, SAM (Gao et al., 2023) | ✓ | × | 12.44 | 48.66 | 6.30 | 98.15 |
| Frequency, FAM | ✓ | × | 12.79 | 46.33 | 6.41 | 97.58 |
| **Temporal, TAM (Ours)** | ✓ | ✓ | **11.19** | **52.77** | **6.27** | **98.55** |

in audio generation. This is likely due to the stronger impact of temporal structure on coherence, perceptual quality, and the dependencies within audio sequences.

## 5.6 Learned Weights of Reducer

We analyzed how our models allocate attention across video frames by visualizing their learned magnitude weights (Fig. 5). The upper figure shows that our model applies varying attention levels, with larger models exhibiting higher weights and more distinct differences. After softmax normalization (bottom figure), consistent trends are observed for various channels, though not all, with model B focusing more on channels 1, 2, and 32. These findings demonstrate that the Reducer effectively captures key features, selectively updating weights to prioritize relevant ones for audio generation. More experiments of Reducer choices are available in the Appendix A.3.

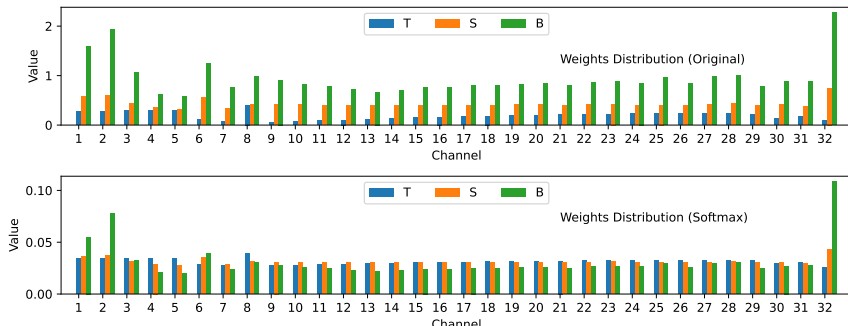

Figure 5: **Learned Weights of Reducer.** Comparison of our three models.

## 5.7 SCALABILITY

We evaluate the scalability of `MDSGen`, the first to explore ViT-based masked diffusion models for vision-guided audio generation. Results in Tab. 8 show that increasing the model size from T to B improves all metrics. However, further scaling to the L model leads to a performance drop, indicating potential overfitting at larger sizes.

Table 8: **Scalability.** We ablate four variants Tiny (T), Small (S), Base (B), and Large (L).

| Model Config. | FID↓ | IS↑ | KL↓ | Align. Acc.↑ | #Params | #Layers | Dim. | #Heads |
|---|---|---|---|---|---|---|---|---|
| **MDSGen-T** | 13.93 | 39.24 | 6.17 | 97.9 | **5M** | 12 | 192 | 3 |
| **MDSGen-S** | 12.92 | 44.38 | 6.29 | 98.3 | 33M | 12 | 384 | 6 |
| **MDSGen-B** | **11.19** | **52.77** | **6.27** | **98.6** | 131M | 12 | 768 | 12 |
| **MDSGen-L** | 12.68 | 49.53 | 6.56 | 97.9 | 461M | 24 | 1024 | 16 |

## 5.8 SAMPLING TOOLS

**Sampling Method.** We employ DPM-Solver (Lu et al., 2022) with 25 steps for sampling during inference. We find that increasing from 25 to 50 steps with dynamic classifier-free guidance (Gao et al., 2023) can slightly improve the performance. We used CFG = 5 and power scaling $\alpha = 0.01$ for the optimal setting. **Classifier-Guidance (CG).** We found that combining CFG and CG slightly improves alignment accuracy and KL, consistent with (Luo et al., 2023), but has no impact on other metrics (Tab. 9), which differs from their findings. A thorough investigation of network architecture and additional datasets is needed to assess the complementary effects of CFG and CG, which is beyond the scope of our work.

Table 9: **CFG and CG.** We examine the effect of classifier-free guidance and classifier guidance. Results are shown with model `MDSGen`-B. Gray indicates the default.

| Setup | FID↓ | IS↑ | KL↓ | Align. Acc.↑ |
|---|---|---|---|---|
| No Guidance | 16.50 | 23.54 | 6.85 | 84.1 |
| CFG | **11.19** | **52.77** | 6.27 | 98.6 |
| CFG + CG | 11.25 | 51.48 | **6.24** | **98.8** |

## 5.9 COMPARE THE EFFICIENCY

We evaluated inference time, parameter count, and memory usage on a single A100 GPU (80GB) with batch size 1. Tab. 10 shows our method is significantly faster, uses fewer parameters, and consumes less memory than existing methods. Specifically, Diff-Foley (860M) achieves 93.9% alignment accuracy with a 0.36s inference time, while our `MDSGen-T` (5M) reaches

Table 10: **Efficiency Comparison.** Our approach is simple and highly efficient across all metrics, with superior alignment accuracy compared to existing methods.

| Method | Time↓ | Mem. Use↓ | #Params↓ | Align. Acc.↑ |
|---|---|---|---|---|
| Im2Wav (Sheffer & Adi, 2023) | 6.41s | 1684M | 448M | 67.4 |
| See and Hear (Xing et al., 2024) | 18.25s | 14466M | 1280M | 58.1 |
| FoleyCrafter (Zhang et al., 2024) | 2.96s | 12908M | 1252M | 83.5 |
| Diff-Foley (Luo et al., 2023) | 0.36s | 5228M | 860M | 93.9 |
| **MDSGen-T (Ours)** | **0.01s** | **1406M** | **5M** | **97.9** |
| **MDSGen-S (Ours)** | 0.02s | 1508M | 33M | 98.3 |
| **MDSGen-B (Ours)** | 0.05s | 2132M | 131M | 98.6 |

97.9% in just 0.01s, 36× faster and 371% more memory efficient. Our larger model `MDSGen-B` (131M) improves accuracy to 98.6%, still 7.2× faster and 245% more memory efficient than Diff-Foley. Compared to FoleyCrafter and See and Hear, `MDSGen-T` is 296× and 1825× faster, respectively, while being 10× more memory efficient. We provide additional details on training efficiency in the Appendix, further emphasizing the remarkable efficiency of our approach.

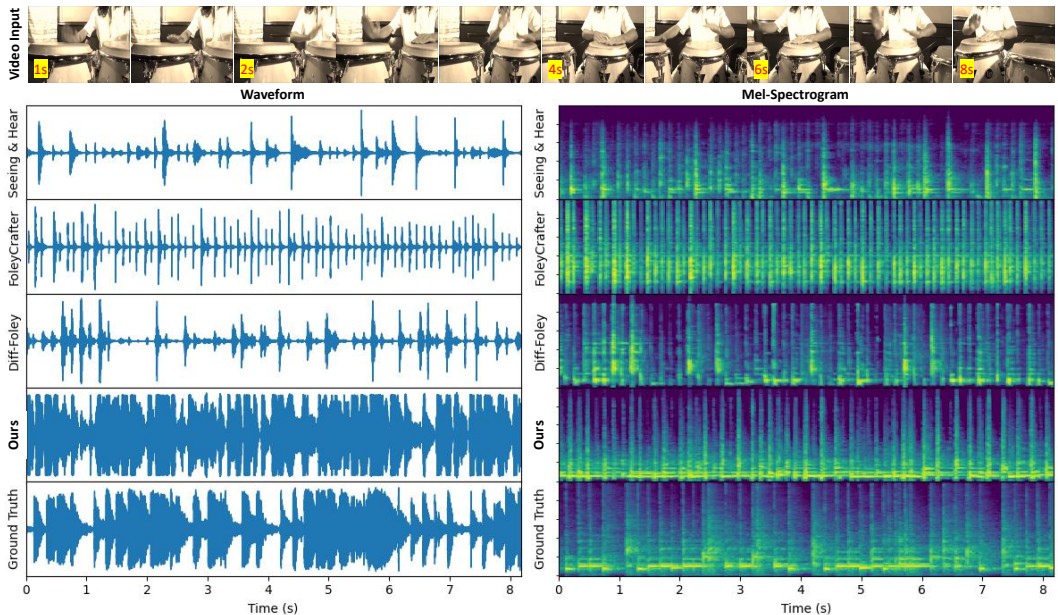

Figure 6: **Waveform and Mel-Spectrogram Comparison.** Sample is taken from the test set of the VGGSound dataset. Our model generates sound more closely aligned with the ground truth than existing methods. **The video of a woman playing drum by hand**, demo file "0NIE-eDk92M_000029.wav" is available in the supplementary material.

## 5.10 VISUALIZATIONS

We visualize different approaches using test video samples from the VGGSound dataset. As shown in Fig. 6, our method generates mel-spectrograms that closely match the ground truth (right figure), with even clearer distinctions observable in the waveform (left figure). Additional demo samples, along with WAV files for convenient listening and their visualizations, are included in the Appendix and supplementary material.

## 6 CONCLUSIONS

This work presents a novel, scalable, and highly efficient framework for video-guided audio generation. Leveraging Diffusion Transformers, we introduced an innovative masking strategy that enhances the model's ability to capture temporal dynamics in audio, leading to significant performance gains. To address redundant video features, we introduced a Reducer module to eliminate unnecessary information. Extensive experiments and detailed analyses demonstrate that our model achieves fast training and inference times, uses minimal parameters, and delivers superior performance across multiple metrics, setting a new benchmark in the field.

### LIMITATIONS AND FUTURE WORKS

Our method offers fast inference, efficient parameter usage, and low memory consumption, while achieving top performance in alignment accuracy and IS score. However, there are some limitations. First, like other diffusion models, it requires multiple sampling steps during generation. Second, while the VGGSound dataset is suitable for this study, its size may not fully leverage the potential of our approach. Third, the current design is constrained to a fixed video length of 8.2 seconds. In the future, we aim to incorporate recent advancements in single-step diffusion techniques to address this limitation. Additionally, although video collection from online sources is becoming more feasible, it remains time-consuming and storage-intensive, which may be challenging for individual researchers. We plan to explore the potential of 1D VAEs for further improvement and to address the fixed-length constraint in future work.

## ACKNOWLEDGMENT

This work was supported by the Institute for Information & Communications Technology Planning & Evaluation (IITP) grant funded by the Korea government (MSIT) (No. RS-2021-II211381, Development of Causal AI through Video Understanding and Reinforcement Learning, and Its Applications to Real Environments) and (No. RS-2022-II220184, Development and Study of AI Technologies to Inexpensively Conform to Evolving Policy on Ethics).

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

# A APPENDIX

## A.1 REDUCER DETAILS

We show the simple design of our Reducer that can help to obtain global information while retaining local information:

- **Input:** Video feature $V \in \mathbb{R}^{32 \times 512}$

- **Output:** feature vector $v \in \mathbb{R}^{1 \times 768}$

Fig. 7 illustrates the details of the proposed Reducer architecture with two layers: the fully connected layer captures the local information of the video, and the second layer (1x1 conv) extracts the global information. Specifically, after the initial layer with fully connected weights, each dimension component (position) from 1 to 768 in the output vector $u_1$ contains the whole vector $v_1$ (local information of video). Consequently, in the next layer, each component of the final vector $v \in \mathbb{R}^{1 \times 768}$ captures both local and global information from the original 32x512 video information. This ensures that the final vector provides comprehensive information for the subsequent audio generation process. This lightweight vector significantly reduces the burden of the DiT process.

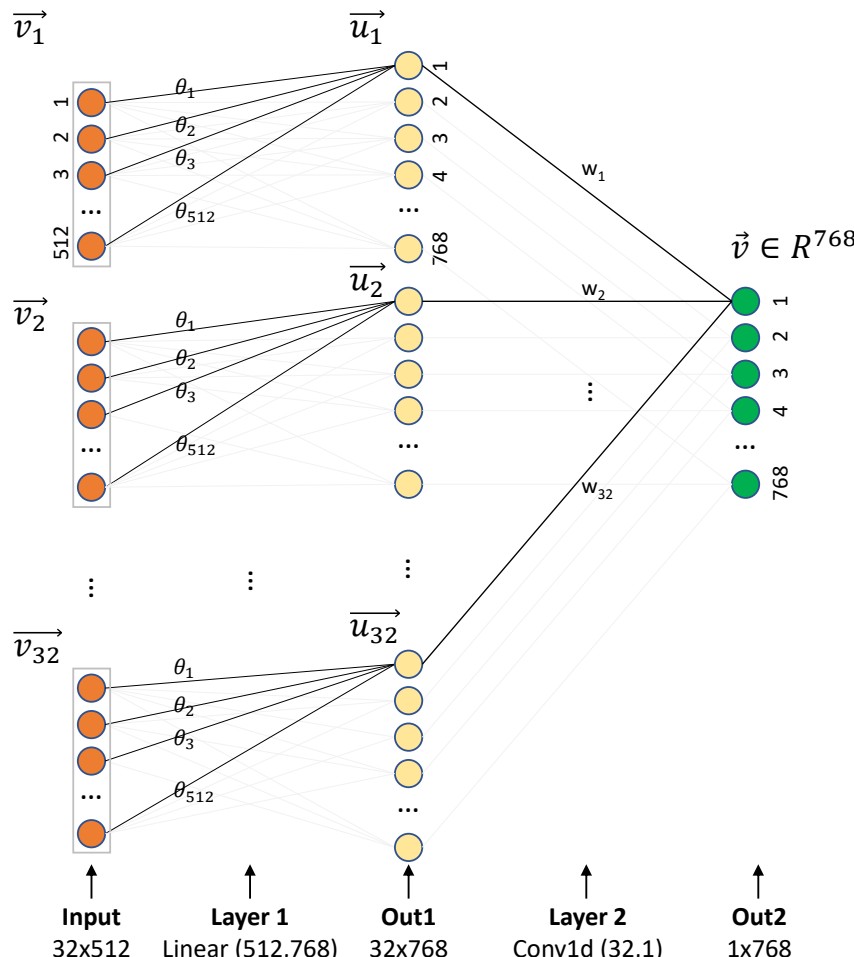

Figure 7: **Reducer Architecture.** It includes a linear layer Linear(512,768) and a 1x1 conv layer Conv1d(32,1) that helps to retain local information while reducing dimension.

## A.2 Overfitting Phenomenon with Redundant Features

We observe that the model becomes quickly overfitted if using redundant features in Fig. 8. By contrast, our Reducer helps to mitigate such redundant features and we can see that the test accuracy remains quite a close gap with train accuracy from 100k steps to 500k steps.

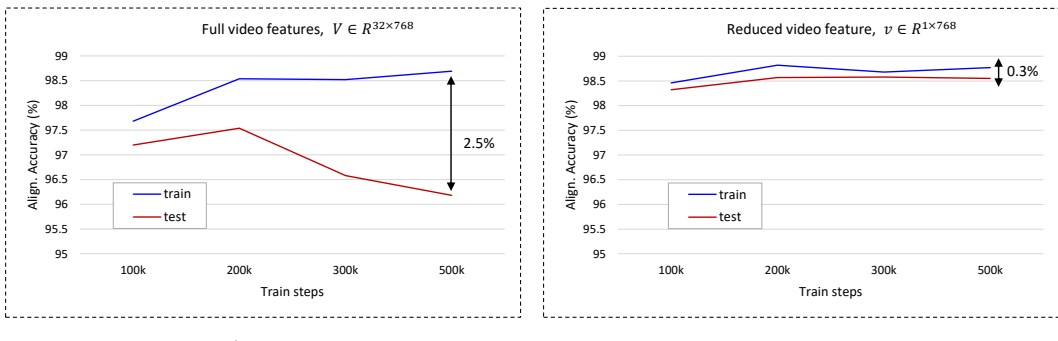

Figure 8: **Overfitting with Redundant Features.** We see that the redundant features show a bigger gap of overfitting where the test and train accuracy gap becomes larger.

## A.3 Reducer Choices

We conducted an ablation study on the choice of the Reducer using three approaches: 1) Naive average pooling, 2) Attention pooling, and 3) Learnable weights. For this study, we used the MDSGen-B model trained for 300k steps. The results, shown in Tab. 11, indicate that all pooling methods achieve competitive performance with comparable alignment accuracy (Align. Acc). However, Learnable Weights yield the highest inception score (IS) at 7 points and slightly outperform in terms of FID. Meanwhile, Attention Pooling achieves the best KL metric. We hypothesize that Attention Pooling

Table 11: **Reducer Vector Pooling Choice.** Model MDSGen-B is trained for 300k steps

| Pooling Method | FID$\downarrow$ | IS $\uparrow$ | KL $\downarrow$ | Align. Acc.$\uparrow$ |
|---|---|---|---|---|
| Naive Average Pooling | 12.5594 | 39.41122 | 6.1750 | 0.9848 |
| Attention Pooling | 12.2704 | 39.0696 | **6.0819** | 0.9847 |
| **Learnable Weight (default)** | **12.1534** | **46.3301** | 6.3066 | **0.9858** |

performs better on the KL metric because its adaptive weights focus on the most relevant features in the input, enabling a more precise reconstruction of the latent distribution and, consequently, better KL divergence minimization. On the other hand, the Learnable Weights method performs best on the inception score because it directly optimizes the contribution of each dimension, tailoring the representation for the final task. This flexibility allows the model to capture both global and local information more effectively, leading to improved perceptual quality as reflected in the IS metric.

Learnable weights can indeed be considered a form of adaptive weighting since the weights are optimized during training and dynamically adjusted based on the data and task requirements. The distinction lies in the mechanism:

- Attention pooling calculates adaptive weights based on the input features themselves (using attention scores). This is data-dependent and can adapt to specific patterns in the input at each forward pass.

- Learnable weights, on the other hand, are parameterized and optimized during training, making them adaptable over time but not directly dependent on the input features in real time.

So while both methods involve adaptivity, attention pooling adapts dynamically per input, whereas learnable weights are statically optimized across the dataset. Naive average pooling is less effective compared to attention pooling and learnable weights because it assigns equal importance to all input

features, regardless of their relevance to the task. This uniform weighting lacks the ability to focus on critical features or filter out irrelevant ones, which can dilute the quality of the pooled representation.

## A.4 CHOICE OF DECODER LAYERS

We compared the number of decoder layers and showed that $N_2 = 4$ gives better results on multiple metrics compared to $N_2 = 2$ as in the below table:

Table 12: **Choice of decoder layer.** Model MDSGen-B is trained for 300k steps

| Decoder | FID↓ | IS↑ | KL↓ | Align. Acc.↑ |
|---|---|---|---|---|
| $N_2 = 2$ | 12.4602 | **47.9981** | 6.4344 | 0.9823 |
| $N_2 = 4$ (**default**) | **12.1534** | 46.3301 | **6.3066** | **0.9858** |

## A.5 TRAINING COST

We leveraged the pre-trained VAE encoder and decoder from Stable Diffusion (Rombach et al., 2022), keeping them frozen during training and inference, similar to Diff-Foley. Our training utilized a single A100 GPU (80GB) with a batch size 64. The B-model (131M) is projected to take 4 days for 500k iterations, while the S-model (33M) and T-model (5M) are expected to finish in 3.3 and 2.8 days, respectively.

In comparison, the second-best method, the Diff-Foley approach (860M model) required 8 A100 GPUs with a batch size of 1760, completing 24.4k steps in 60 hours (2.5 days) (Luo et al., 2023). If scaled to a single A100 GPU, Diff-Foley would need at least 20 days more than a fifth of the time of our method, demonstrating the superior efficiency and significantly lower training costs of our approach (see Tab. 13).

Table 13: **Training Comparison.** Estimated for a single A100 GPU training. Our approach is simple and highly efficient compared to the second-best method Diff-Foley which used the heavy backbone of Stable Diffusion with 860M.

| Method | #Training cost↓ | Align. Acc.↑ |
|---|---|---|
| Diff-Foley (860M) (Luo et al., 2023) | 20 days | 93.9 |
| **MDSGen-T, 5M (Ours)** | **2.8 days** | **97.9** |
| **MDSGen-S, 33M (Ours)** | **3.3 days** | **98.3** |
| **MDSGen-B, 131M (Ours)** | **4.1 days** | **98.6** |

## A.6 MORE SETUPS

We apply classifier-free guidance during training by randomly setting $\vec{v}$ to zero with a 10% probability. Models are trained for 500k steps to ensure optimal convergence. The exponential moving average of the model weight is set to 0.9999, otherwise, settings are the same as default DiT (Peebles & Xie, 2023). No video augmentation is used; instead, we pre-extract and save lightweight video features for faster training. We also use a ratio of $\eta_m = 0.3$ by default for masking the temporal set of tokens. Our code will be made publicly available.

For experiments involving classifier guidance, we utilized the classifier trained by Diff-Foley, adjusting the optimal CG value to 2.0, compared to 50 in their framework. To evaluate alignment accuracy, we used their trained classifier to assess our generated audio. We also reproduced Diff-Foley's performance using their published checkpoint, with results closely matching their reported metrics.

The slight variation may stem from differences in the VGGSound test set, as we download videos from several months to one year after their experiments, during which some original YouTube links may have been removed, causing potential mismatches.

## A.7 COMPARING MORE VISUALIZATIONS

**Gradcam Visualization of localization on VGGSound and Flickr-SoundNet datasets.**

We used the generated audio to perform the sound source localization on each frame of the video and image using the pre-trained model of EZ-VSL (Mo & Morgado, 2022). As shown in Fig. 9 and Fig. 11, our method provides a more accurate attention map.

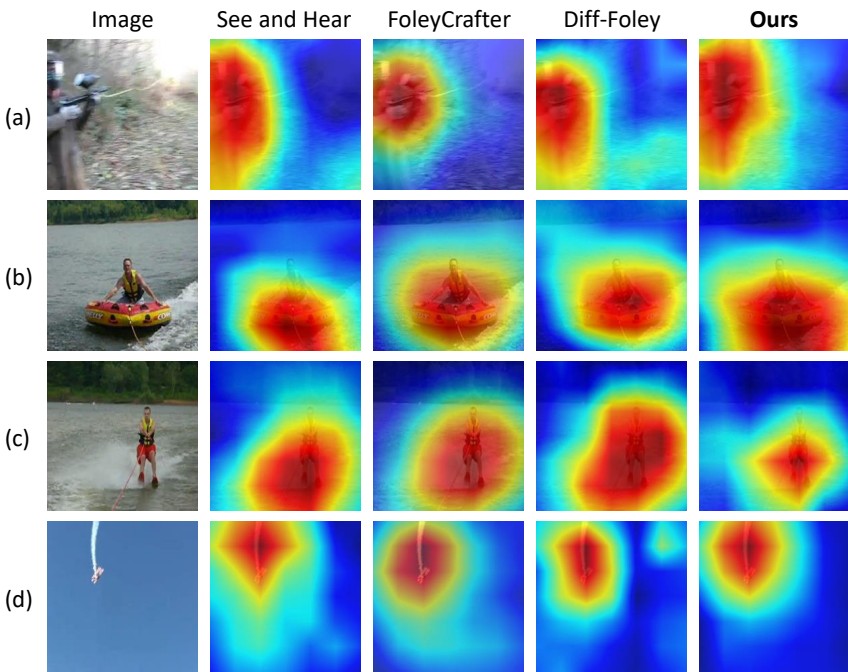

Figure 9: **Attention map Flickr-SoundNet dataset.** Our best model (MDSGen-B) generated the sound that contain information that help localize the sound source more accurately compared to existing approaches.

**More generated audio comparison with state-of-the-art approaches on VGGSound test set.**

We also provide more samples in the supplementary and their visualizations in Fig. 12, Fig. 13, Fig. 15, Fig. 16, and Fig. 14. As shown in these figures and listened to by authors, our generated audio is much more reasonable than others. We refer readers to examine the quality of generated audio in the submitted supplementary materials.

### A.8 CHANNEL SELECTION FROM RGB FOR MEL-SPECTROGRAM

We provide additional statistics of various generated audio samples, highlighting the differing characteristics of the VAE decoder outputs in Fig. 17 and Fig. 18. As shown, although the VAE encoder input consists of three identical channels, the generated outputs display distinct distributions across each channel (left figures), even though these differences are imperceptible to the human eye (right figures). This behavior stems from the fact that the VAE encoder and decoder in Stable Diffusion (Rombach et al., 2022) are trained exclusively for image data, where the R, G, and B channels inherently carry different information.

Because this model is applied directly to audio data without adaptation, there is no constraint ensuring the R, G, and B channels remain identical in the generated audio. Developing a method to adaptively select or combine these channels when constructing the final Mel-spectrogram could be a promising avenue for improving the quality of the generated audio.

Each R, G, and B channel exhibits distinct characteristics, as noted in previous research (Xu et al., 2017). In the VAE encoder, we replicate the gray mel-spectrogram across three identical channels, but the VAE decoder does not enforce channel consistency. Our analysis shows that

Table 14: **Channel selection for mel-spectrogram.** Gray indicates the default.

| Channel | FID↓ | IS↑ | KL↓ | Align. Acc.↑ (%) |
|---|---|---|---|---|
| $R\ (i=0)$ | 11.40 | 51.54 | 6.29 | 98.51 |
| $G\ (i=1)$ | **11.19** | **52.77** | **6.27** | 98.55 |
| $B\ (i=2)$ | 11.23 | 52.32 | **6.27** | **98.56** |
| $\frac{1}{3}\sum_{r\in\{R,G,B\}} r$ | 11.29 | 52.27 | 6.28 | 98.54 |

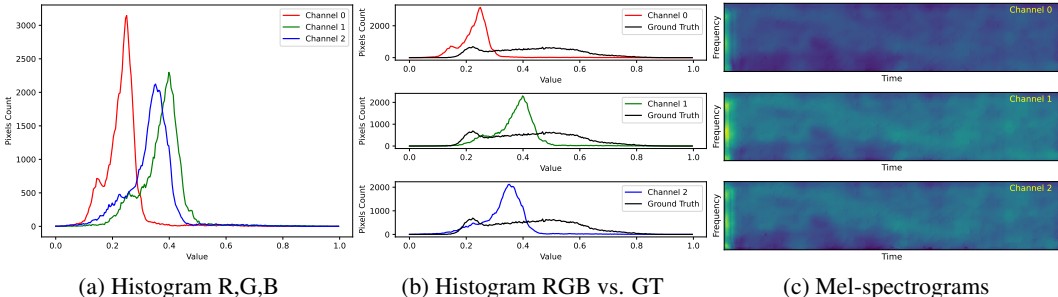

(a) Histogram R,G,B  (b) Histogram RGB vs. GT  (c) Mel-spectrograms

Figure 10: Histogram and mel-spectrogram comparison of three channels of VAE output.

the RGB output $\widehat{\mathbf{X}}_{RGB}$ retains unique statistical differences across channels (see Fig. 10), influencing their contributions to the final mel-spectrogram and waveform. In contrast to Diff-Foley, which uses only the R channel ($\widehat{\mathbf{X}}_{RGB}[0,:,:]$) for the final mel-spectrogram, we find the G channel ($\widehat{\mathbf{X}}_{RGB}[1,:,:]$) to be optimal (see Tab. 14). Fig. 10 shows that the histograms of the three VAE output channels display significant differences, with the G and B channels aligning closely with the ground truth distribution.

Notably, while the resulting mel-spectrograms (right figures) seem visually indistinguishable, the histograms highlight their differences. This emphasizes the importance of considering each channel's statistics in generating the final mel-spectrogram $\widehat{\mathbf{X}}$, with more comparisons in the Appendix.

## A.9 Mask Ratio Ablation

Tab. 15 shows that while a higher masking ratio maintains high alignment accuracy, it leads to declines in other metrics. This occurs because the transformer models prioritize audio token reconstruction over the primary generation task, resulting in worsened FID, IS, and KL scores.

Table 15: **Masking Strategy for Audio Synthesis.** Comparison of different ways to train diffusion transformer-based models. Masking on temporal audio gives the best performance.

| Masking Ratio | FID↓ | IS↑ | KL↓ | Align. Acc.↑ (%) |
|---|---|---|---|---|
| 70% | 12.85 | 44.42 | 6.38 | 97.86 |
| 50% | 12.39 | 46.77 | 6.38 | 98.43 |
| 30% | **11.19** | **52.77** | **6.27** | **98.55** |

## A.10 More Directions

Our approach can inspire applications of masked diffusion models across various domains such as object detection (Vu et al., 2019), video question answering (Kim et al., 2020), image super-resolution (Niu et al., 2024a;b), and speech processing. These models can also be combined with self-supervised learning techniques (He et al., 2022; Pham et al., 2021; 2022b; 2023; 2022a) to enhance representation learning for diverse generative tasks. With the potential of diffusion transformers for conditional learning, we anticipate further exploration of their capabilities in speech processing (Jung et al., 2022; 2020; Trung & Yoo, 2019), data augmentation (Lee et al., 2020), VQA (Kim et al., 2020), visual detection (Vu et al., 2019), and super-resolution (Niu et al., 2023; 2024a;b). Our work demonstrates the potential of Masked Diffusion Transformers for audio generation while also highlighting their applicability to various other tasks, including image editing (Koo et al., 2024; Song et al., 2024; Yoon et al., 2024) and personalized image generation using flow-based methods (Pham et al., 2024; Lipman et al., 2022).

## B VAE Choices for Melspectrogram Reconstruction

We conducted an experiment to assess the impact of the VAE in our overall framework. Specifically, we compare the often used Stable Diffusion VAE which is used in Diff-Foley (Luo et al., 2023) and

Action2Sound (Chen et al., 2024) and compared with the case when we employ the AudioLDM VAE to replace Stable Diffusion VAE. The results are shown in the below tables. Several key findings are observed. Below, we provide a side-by-side comparison highlighting the performance differences between the image VAE and the AudioLDM VAE across key metrics such as FAD, IS, KL, and Alignment Accuracy, and efficiency.

Part 1. Performance Comparison. We find that although the AudioLDM VAE achieves better results in FAD and KL, with a comparable FID score, its performance in Alignment Accuracy and IS is not as strong as that of the image VAE. Our smallest model, MDSGen-T (5M), achieves a state-of-the-art FAD score of 2.21, surpassing existing video-to-audio methods (the FAD of FoleyCrafter is 2.45). The larger model, MDSGen-B (131M), further improves the FAD to 1.34. Note that, we reported FAD 2.16 of our MDSGen in the previous response from checkpoint 500k training steps with image VAE. The detailed results are summarized in Tab. 1 below (300k steps):

Table 16: **Performance Comparison AudioVAE**: Image VAE vs. AudioLDM VAE

| Model config. (Ours) | VAE choice | FID↓ | IS↑ | KL↓ | Align. Acc.↑ (%) | FAD↓ |
|---|---|---|---|---|---|---|
| MDSGen-B (131M) | ImageVAE (Rombach et al., 2022) | 12.15 | **46.41** | 6.31 | **98.58** | 2.36 |
| | AudioLDM VAE (Liu et al., 2023) | **11.83** | 30.37 | 6.03 | 96.52 | **1.34** |
| MDSGen-S (33M) | ImageVAE (Rombach et al., 2022) | 12.88 | **40.39** | 6.21 | **98.25** | 2.75 |
| | AudioLDM VAE (Liu et al., 2023) | 13.14 | 26.05 | 5.83 | 95.63 | **1.66** |
| MDSGen-Tiny (5M) | ImageVAE (Rombach et al., 2022) | 14.78 | **33.23** | 6.23 | **97.62** | 3.69 |
| | AudioLDM VAE (Liu et al., 2023) | 15.48 | 21.99 | 5.58 | 95.53 | 2.21 |

Part 2. Efficiency Comparison. We observed a slight difference in efficiency between the two VAEs, primarily due to variations in input dimensions and output characteristics. For instance, the AudioLDM VAE produces 816 latent tokens, which is 3.1x longer than the 256 tokens generated by the image VAE. As a result, models using the AudioLDM VAE tend to be slower and require more memory. However, it's important to note that all variants of our model remain considerably more efficient than existing baseline methods. The details are shown in Tab. 2 below:

Table 17: **Efficiency Comparison**: Image VAE vs. AudioLDM VAE

| VAE | MelSpec. Res. | Latent | #Tokens | Models | #Prams. (M)↓ | Infer. Time (s).↓ | #Memory (M)↓ |
|---|---|---|---|---|---|---|---|
| Image VAE | $128 \times 512$ | $16 \times 64$ | 256 | MDSGen-T | 5M | **0.01** | **1406** |
| | | | | MDSGen-S | 33M | **0.02** | **1508** |
| | | | | MDSGen-B | 131M | **0.05** | **2132** |
| AudioLDM VAE | $64 \times 816$ | $16 \times 204$ | 816 | MDSGen-T | 5M | 0.058 | 3087 |
| | | | | MDSGen-S | 33M | 0.114 | 5383 |
| | | | | MDSGen-B | 131M | 0.243 | 10165 |

**VAE Analysis** **Insight 1**. Compared to existing video-to-audio methods, both VAEs show that MDSGen achieves state-of-the-art performance in FAD (1.34–2.16) and Alignment Accuracy (96.5%–98.5%). While the Image VAE outperforms the AudioLDM VAE by about 2% in Alignment Accuracy, it falls behind in FAD (2.16 vs. 1.34), indicating that AudioLDM produces higher-quality audio. This discrepancy in alignment accuracy and IS may be attributed to several factors:

Alignment Classifier Training: The alignment classifier is trained on 128×512 spectrograms, whereas AudioLDM uses 64×816 mel-spectrograms, inherently favoring features from the Image VAE, which aligns more closely with its input format. Domain-Specific Design: AudioLDM is optimized for audio reconstruction, resulting in features less visually aligned with the mel-spectrograms expected by the classifier. Resolutions: Image VAEs operate at higher spatial resolutions (128×512) and preserve finer, visually coherent patterns, leading to higher IS scores, while AudioLDM's lower resolution (64×816) focuses on acoustic fidelity rather than visual consistency. Overall, we hypothesize that the Image VAE performs better in IS and alignment accuracy due to its compatibility with visual-based metrics and the image-like nature of spectrograms. In contrast, Audio VAEs prioritize audio quality, creating a trade-off between perceptual fidelity and visual alignment in video-to-audio tasks.

**Insight 2**. Both configurations show that MDSGen is highly efficient, utilizing significantly fewer parameters than existing approaches while maintaining strong performance across key metrics, including FAD, FID, and audio-video alignment accuracy.

**Insight 3**. The AudioLDM VAE improves audio quality over the image VAE, but at the cost of slower inference and slightly lower audio-video content relevance. Overall, our results reveal a trade-off: integrating MDSGen with the image VAE achieves the highest Alignment Accuracy and faster inference, while the AudioLDM VAE delivers superior audio quality (FAD) with slower inference.

**In summary**, our masked diffusion model, MDSGen, is the first to support both image- and audio-based VAEs for the video-to-audio task, offering a well-balanced trade-off between alignment accuracy and high-quality audio, while maintaining exceptional efficiency in parameters, memory usage, and inference speed. In the revised PDF, we have highlighted this in Section 3.5: VAE Choice for Mel-Spectrogram, where we discuss replacing the channel selection used in the RGB VAE. Additionally, we have moved the RGB-related discussion to the Appendix and will include a detailed analysis demonstrating the compatibility of our model with both image and audio VAEs.

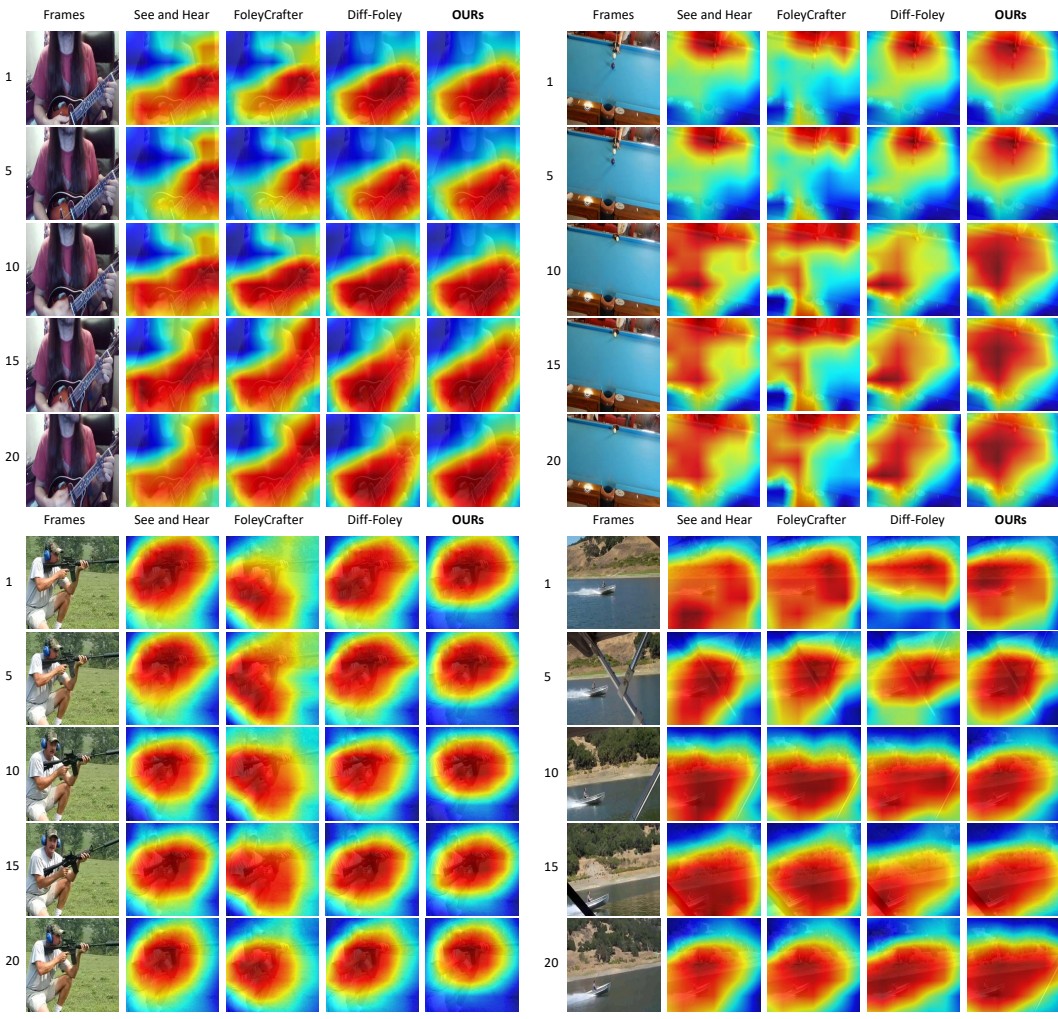

Figure 11: **Attention map VGGSound dataset.** Our best model (MDSGen-B) generated the sound that contain information that help localize the sound source more accurately compared to existing approaches.

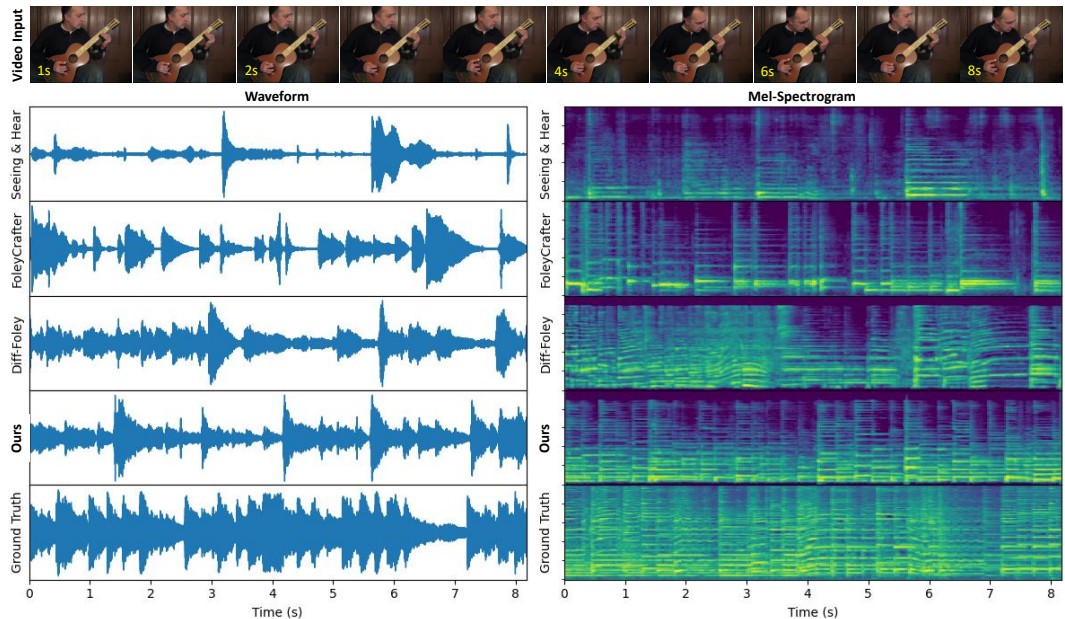

Figure 12: **The video of a man playing guitar solo.** Our best model (MDSGen-B) generated a sound that is closer to GT compared to existing approaches. We refer the reader to the listen file provided in the supplementary for comparison. File "**-lPXTBXa0tE_000030.wav**".

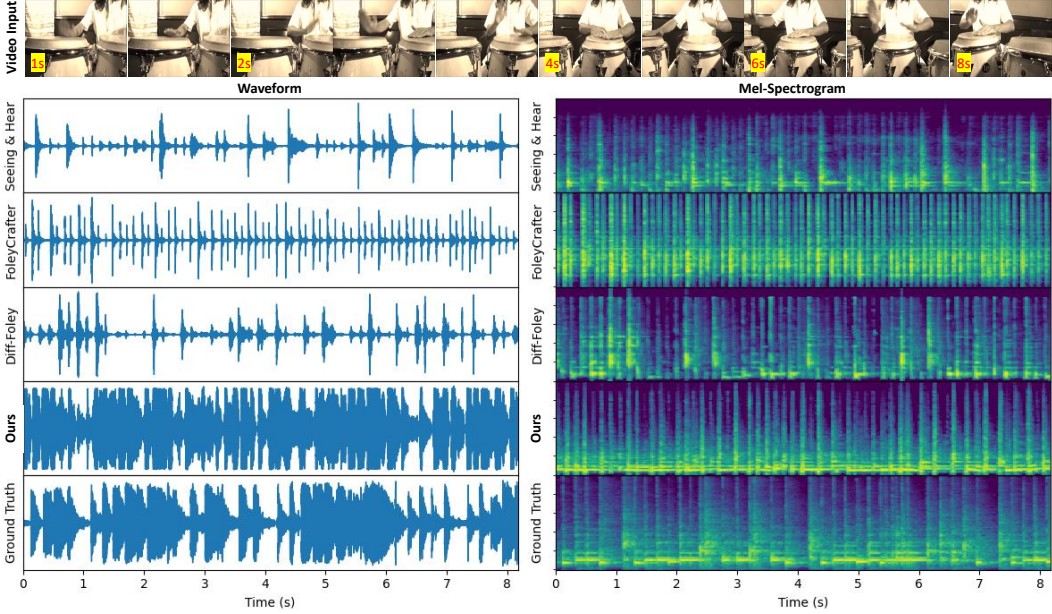

Figure 13: **The video of a woman playing drum by hand.** Our best model (MDSGen-B) generated a sound that is closer to GT compared to existing approaches. We refer the reader to the listen file provided in the supplementary for comparison. File "**0NIE-eDk92M_000029.wav**".

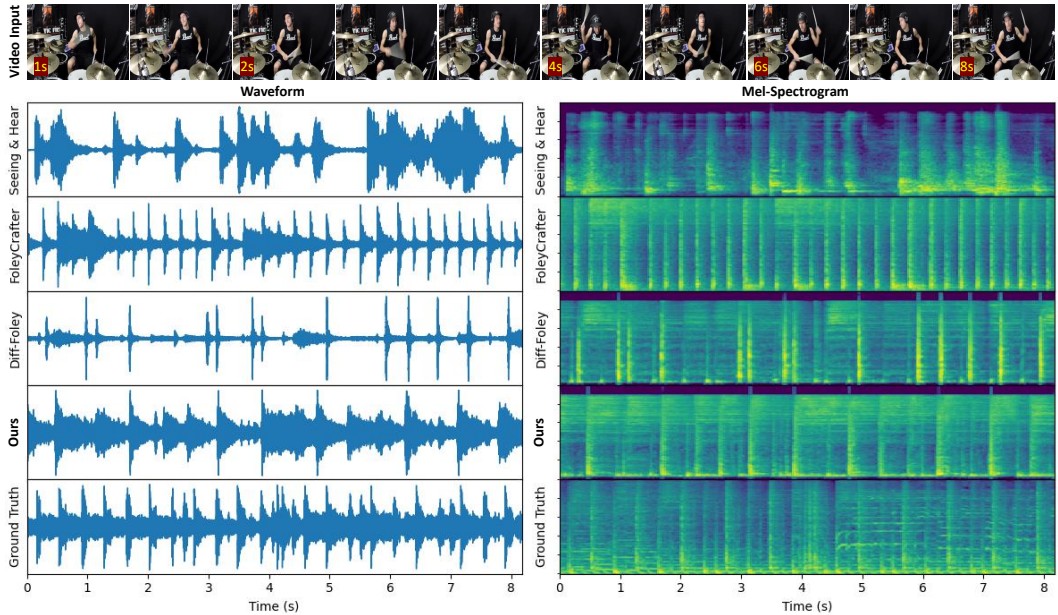

Figure 14: **The video of a guy playing drum by tool.** Our best model (MDSGen-B) generated a sound that is closer to GT compared to existing approaches. We refer the reader to the listen file provided in the supplementary for comparison. File "**-Qowmc0P9ic_000034.wav**".

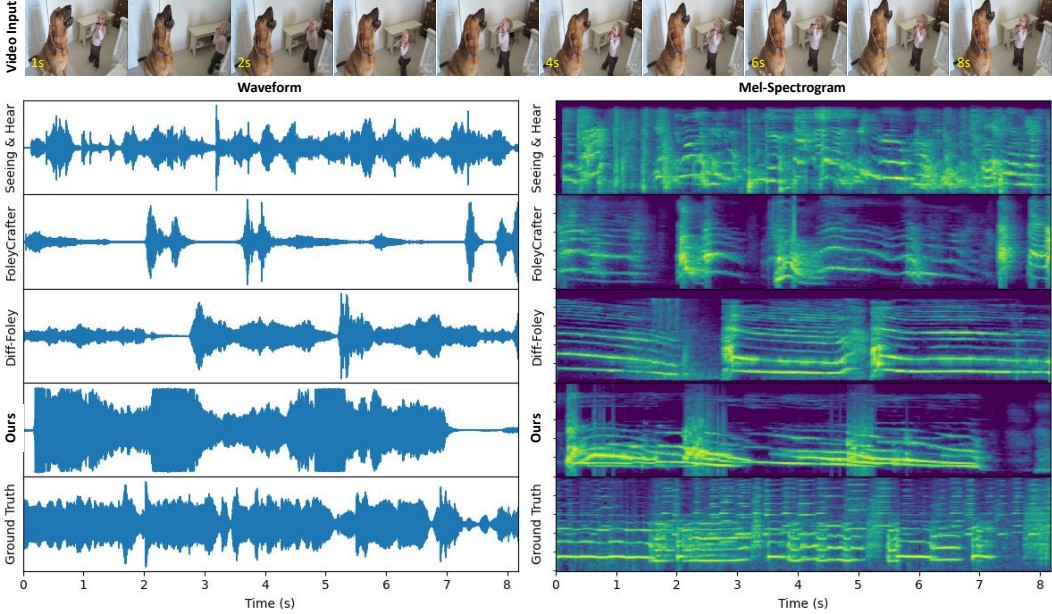

Figure 15: **The video of a dog looks like howling**. Our best model (MDSGen-B) generated a sound closer to GT than existing approaches. We refer the reader to the listen file provided in the supplementary for comparison. File "**2vYkvwD-fkc_000010.wav**".

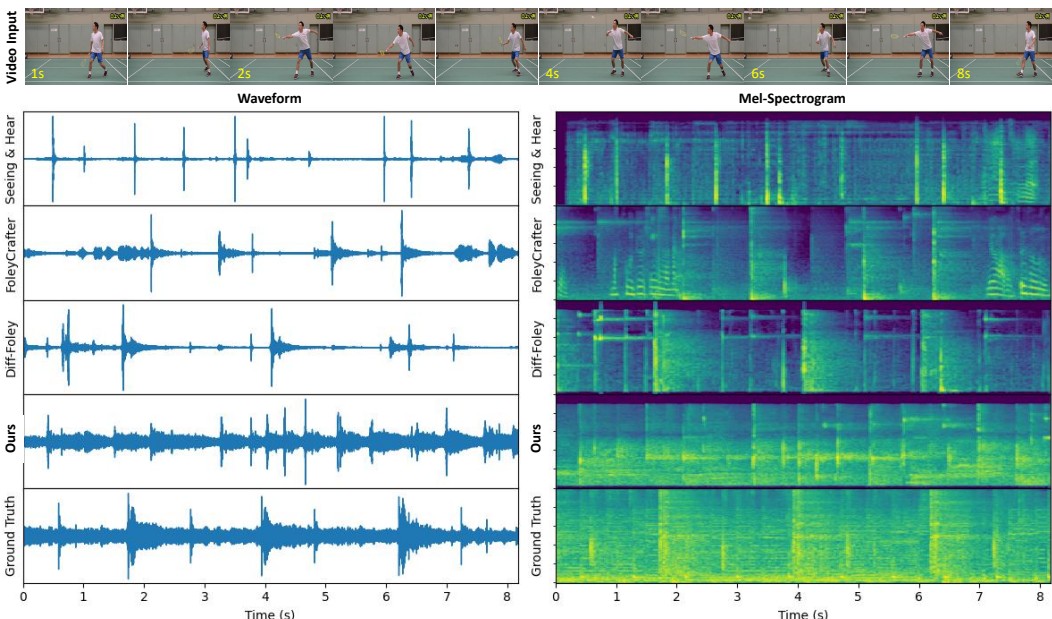

Figure 16: **The video of a guy playing badminton.** Our best model (`MDSGen`-B) generated a sound that is closer to GT compared to existing approaches. We refer the reader to the listen file provided in the supplementary for comparison. File "**-miI_C3At4Y_000104.wav**".

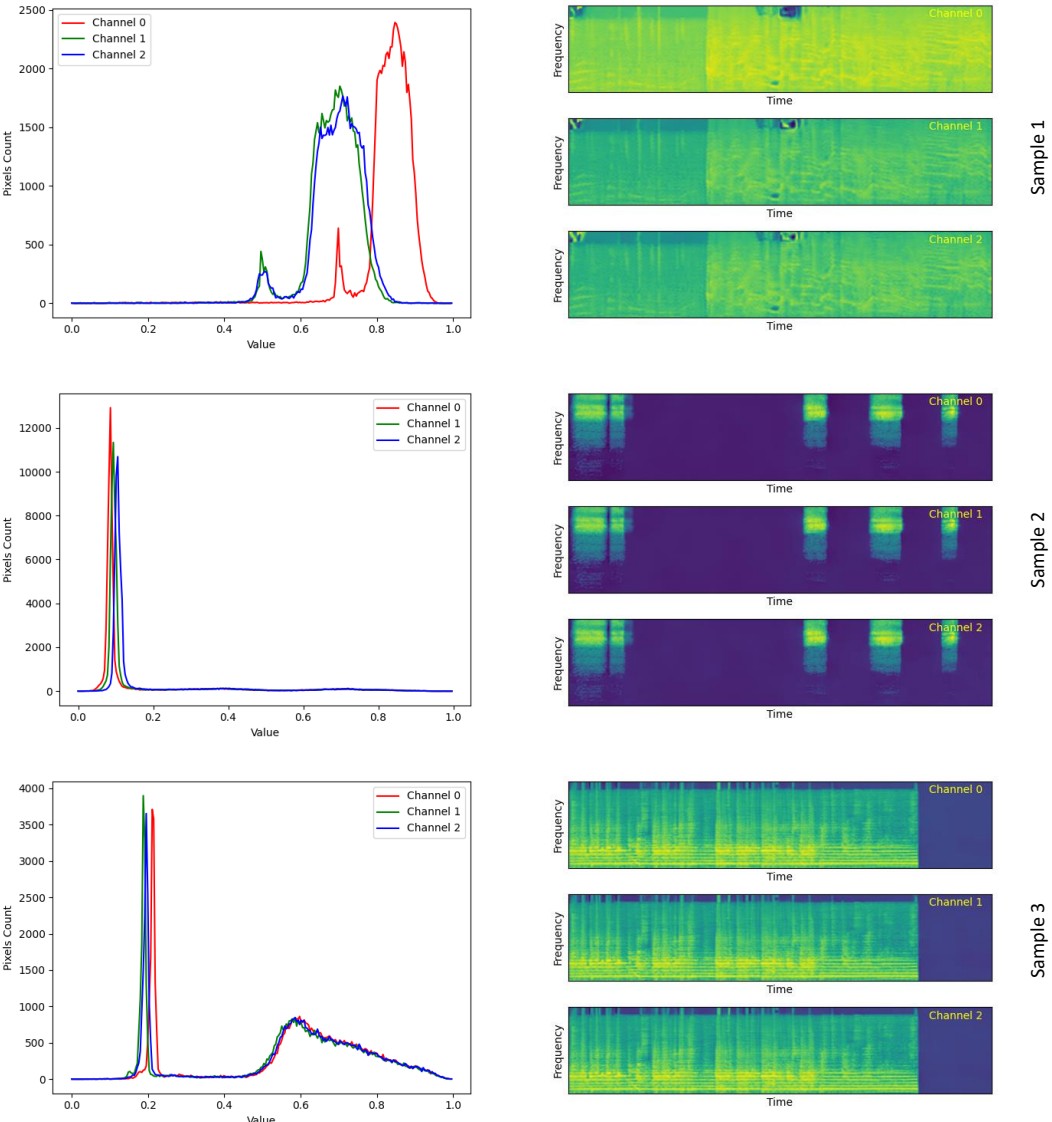

Figure 17: **RGB distribution for Mel-Spectrogram.** We provide more evidential samples that the output of the VAE in the test set yields different characteristics for three channels even though these differences are imperceptible to the human eye (right figures). Interestingly, we find that the first channel (R) always has some different patterns compared to the remained channels (1).

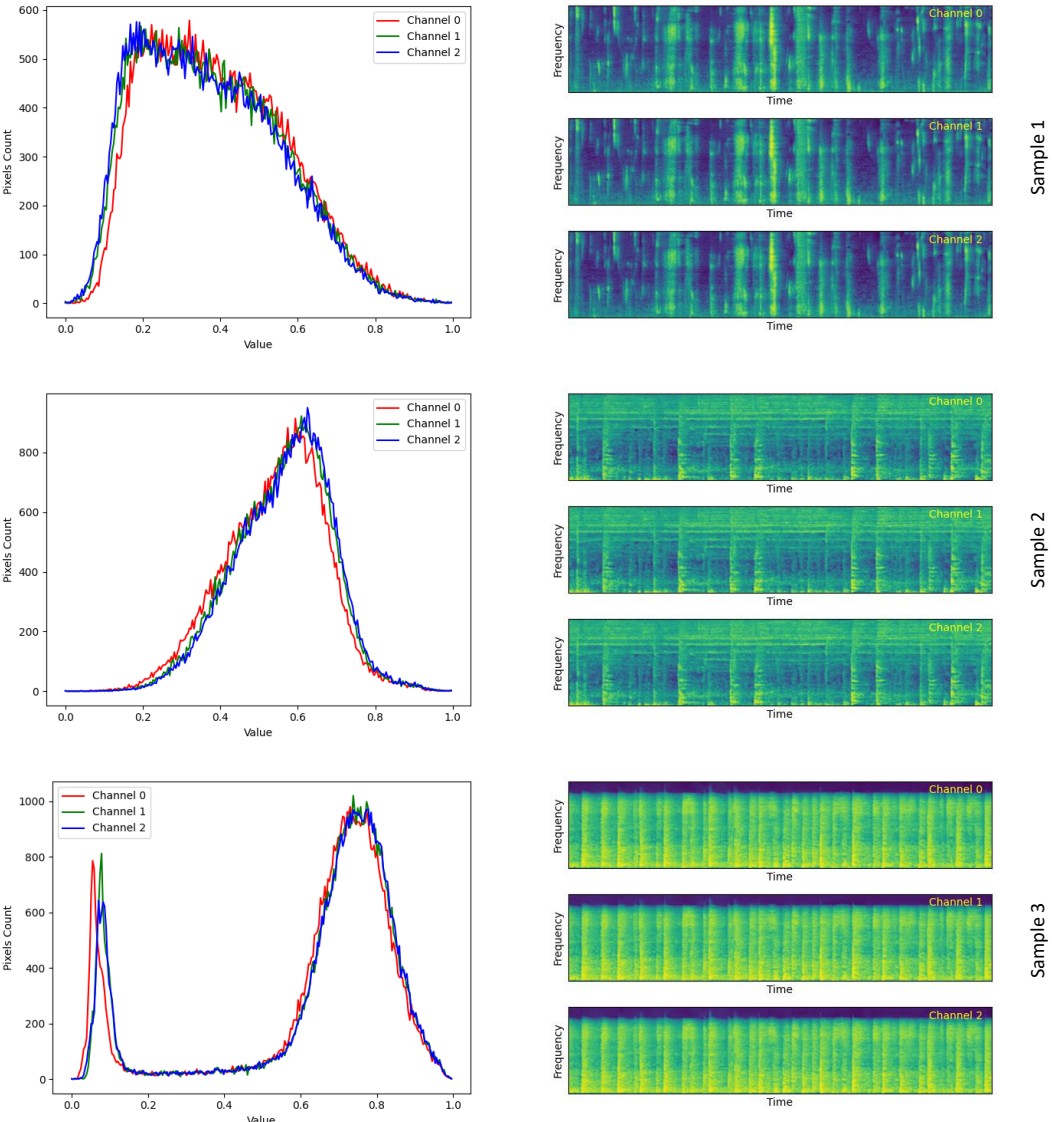

Figure 18: **RGB distribution for Mel-Spectrogram.** We provide more evidential samples that the output of the VAE in the test set yields different characteristics for three channels even though these differences are imperceptible to the human eye (right figures). Interestingly, we find that the first channel (R) always has some different patterns compared to the remained channels (2).

