# OpenReview forum: "MDSGen: Fast and Efficient Masked Diffusion Temporal-Aware Transformers for Open-Domain Sound Generation"
_ICLR.cc/2025/Conference — ICLR 2025 Poster_

### Official Review · Reviewer_mmH9 · 2024-11-02

**Soundness:** 3
**Presentation:** 4
**Contribution:** 4
**Rating:** 6
**Confidence:** 4

**Summary:**

This paper presents MDSGen, an efficient and compact framework for generating open-domain sounds guided by visual input. MDSGen uses masked diffusion transformers instead of the usual U-Net architectures, optimizing for faster speed, lower memory use, and parameter efficiency. Key features include a module to remove redundant video data and a Temporal-Awareness Masking (TAM) strategy, both aimed at efficient, high-quality audio-visual alignment. Tests on the VGGSound and Flickr-SoundNet datasets show that MDSGen achieves strong alignment accuracy with much lower computational demands than larger models.

**Strengths:**

1. MDSGen achieves strong results with a small model size, making it useful for real-time applications. Compared to larger models, it is faster and more memory-efficient.
2. TAM is an interesting approach that focuses on time-based information in audio, aiming to improve alignment by using masking based on temporal patterns rather than spatial patterns (commonly used for images).
3. The paper provides extensive experiments with detailed comparisons against other models. Ablations for each key component further clarify the model’s design choices.

**Weaknesses:**

1. I question the decision to use an image-trained VAE (from Stable Diffusion) rather than an audio-specific VAE, such as those in AudioLDM [1]. An audio-dedicated VAE could better capture the temporal and spectral nuances inherent to sound, which are often lost when treating audio as an image. Relying on an image-based VAE reduces the model’s potential to fully leverage audio-specific features and may affect TAM’s performance.
2. The authors highlight channel selection within the RGB output as a means of optimizing the final mel-spectrogram. While using the G channel showed marginal improvements, I question if relying on such RGB channel selection can sufficiently address the nuances of audio spectrogram representation (similar to 1). A more audio-specific solution that doesn’t require treating spectrograms as RGB images would likely be more consistent with the needs of audio data, as these channels are meant for pixels, not spectral data.
3. I would suggest that the authors conduct a human perceptual study to better assess audio quality. Relying solely on quantitative metrics may not fully capture perceptual quality, as these measures can sometimes be unreliable.
4. In Section 5.4, various masking strategies are explored, and TAM shows a clear improvement over random masking and FAM. However, the reasons for TAM’s superiority are not fully explained. It would be beneficial to discuss why TAM outperforms FAM in this context, particularly since FAM is intuitively suitable for audio data.
5. I noticed a missing citation for SpecAugment [2] and AudioMAE [3], a masking approach relevant to TAM proposed here.

References

[1] Liu et al. AudioLDM: Text-to-Audio Generation with Latent Diffusion Models.

[2] Park et al. SpecAugment: A Simple Data Augmentation Method for Automatic Speech Recognition.

[3] Huang et al. Masked Autoencoders that Listen.

**Questions:**

See weaknesses.

---

> ### Author Response · Authors · 2024-11-20
> **AuthorResponse 1/N**
>
> Dear reviewer mmH9, thank you for recognizing our work and your great comments, we would like to address your concerns one by one as follows.
>
> > **Weakness 1.** I question the decision to use an image-trained VAE (from Stable Diffusion) rather than an audio-specific VAE, such as those in AudioLDM [1]. An audio-dedicated VAE could better capture the temporal and spectral nuances inherent to sound, which are often lost when treating audio as an image. Relying on an image-based VAE reduces the model’s potential to fully leverage audio-specific features and may affect TAM’s performance.
>
> **Re:** We used the VAE from Stable Diffusion, following prior works like Diff-Foley [P1] (NeurIPS 2023), which demonstrated that the VAE in Stable Diffusion effectively reconstructs mel-spectrograms (Table 6 and Fig. 10). This makes it well-suited for leveraging the existing latent diffusion framework to predict mel-spectrogram latents. For a more detailed explanation of our choice of VAE, we refer the reviewer to our response to reviewers Hij3 and GqkQ. While using a VAE specifically designed for audio could potentially enhance our method, our primary focus is on creating a more efficient framework than existing large models like Diff-Foley. To ensure a fair comparison and highlight our method’s advantages, we have retained the same VAE as in their framework. We will explore this in future work.
>
> [1] Diff-Foley: Synchronized Video-to-Audio Synthesis with Latent Diffusion Models, NeurIPS 2023.
>
> > **Weakness 2.** The authors highlight channel selection within the RGB output as a means of optimizing the final mel-spectrogram. While using the G channel showed marginal improvements, I question if relying on such RGB channel selection can sufficiently address the nuances of audio spectrogram representation (similar to 1). A more audio-specific solution that doesn’t require treating spectrograms as RGB images would likely be more consistent with the needs of audio data, as these channels are meant for pixels, not spectral data.
>
> **Re:** We fully agree with the reviewer that an audio-specific VAE would eliminate the need for channel selection. In our work, we maintained consistency with the Diff-Foley baseline, which was the first to apply a diffusion model to video-to-audio. Specifically, we used the same mel-spectrogram (128x512) for training the CAVP video encoder, which is then utilized in the diffusion process. By keeping the same VAE and CAVP, we focused on developing a new, lightweight transformer-based diffusion model rather than relying on their heavier Stable Diffusion backbone. Channel selection is a refinement we introduced to enhance their approach. We plan to explore an audio-specific VAE for further improvement and have included this discussion in our revised paper.

---

> ### Author Response · Authors · 2024-11-20
> **AuthorResponse 2/N**
>
> > **Weakness 3.** I would suggest that the authors conduct a human perceptual study to better assess audio quality. Relying solely on quantitative metrics may not fully capture perceptual quality, as these measures can sometimes be unreliable.
>
> **Re:** We conduct a human evaluation with MOS on two aspects: audio quality (AQ) and video-audio relevance (AV). We report results as follows:
>
> **Table 1. Human evaluation with a subjective listening test**
> | Metric\Method |  See-and-hear |  FoleyCrafter |   Diff-Foley  |  MDSGen-B (Ours)  |  Ground Truth  |
> |:-------------:|:-------------:|:-------------:|:-------------:|:-----------------:|:--------------:|
> |     MOS-AQ $\uparrow$    | 2.68$\pm$0.25 | 3.21$\pm$0.23 | 3.29$\pm$0.24 | **3.66$\pm$0.23** | 4.74 $\pm$0.12 |
> |     MOS-AV $\uparrow$    | 2.95$\pm$0.20 | 3.44$\pm$0.26 | 3.56$\pm$0.23 | **3.76$\pm$0.21** |  4.62$\pm$0.23 |
>
> The results from our human evaluation demonstrate that the waveforms generated by our model outperform those of competing methods, receiving higher scores across the evaluation criteria. Participants consistently rated the audio quality and alignment with the video content more favorably for our generated waveforms, indicating superior perceptual performance.
>
> > **Weakness 4.** In Section 5.4, various masking strategies are explored, and TAM shows a clear improvement over random masking and FAM. However, the reasons for TAM’s superiority are not fully explained. It would be beneficial to discuss why TAM outperforms FAM in this context, particularly since FAM is intuitively suitable for audio data.
>
> **Re:** We can see TAM outperforms the FAM. In audio generation tasks, masking in the temporal dimension generally provides better performance than masking in the frequency dimension likely due to the distinct way temporal structure impacts coherence, perceptual quality, and the inherent dependencies in audio sequences. The possible reasons why temporal masking tends to provide better results than frequency masking are listed as follows:
> + **Temporal:** Audio is fundamentally structured as a time-evolving signal, with temporal continuity playing a key role in maintaining the coherence and natural flow of sound. Masking in the temporal dimension pushes the model to learn dependencies across time steps, capturing transitions and rhythmic patterns crucial for realistic audio generation. This temporal structure is especially important in human perception, as even slight temporal inconsistencies are more noticeable than frequency variations.
> + **Frequency:** Unlike temporal information, frequency information in audio often has redundancy across frames (i.e., similar spectral patterns repeating over time, it is clearly observed in mel-spectrogram image). Masking frequencies can thus lead to less distinctive learning, as the model may focus on repetitive spectral features rather than meaningful temporal dependencies. Temporal masking, on the other hand, minimizes redundancy by promoting a richer contextual understanding of the sequence.
> + **From the perspective of perceptual sensitivity of the human:** Humans are more sensitive to artifacts in the time dimension than in frequency, where minor spectral variations are less perceptible. Temporal masking therefore aligns with human auditory perception, helping the model to learn features that ensure smooth, realistic audio playback.
>
> We updated an analysis of TAM's advantage over FAM in our revised paper in red lines.
>
> > **Weakness 5.** I noticed a missing citation for SpecAugment [2] and AudioMAE [3], a masking approach relevant to TAM proposed here.
>
> **Re:** We have incorporated SpecAugment and AudioMAE into the revised version of the paper, providing a detailed explanation of their similarities and differences compared to our work.
>
> Specifically, AudioMAE and SpecAugment share a commonality with our approach in their use of masking for audio data. However, there are distinctions:
> + Masking Domain: They apply masking directly in the pixel space of mel-spectrograms, whereas our method performs masking in the latent space of a VAE.
> + Objective: Their methods focus on masking for representation learning in downstream recognition tasks, while our approach leverages masking for audio generation.
> + Data Context: Their masking is applied to clean mel-spectrograms, whereas our work incorporates masking within noisy latent variables in the diffusion framework.

---

> > ### Comment · Reviewer_mmH9 · 2024-11-25
> >
> > Thank you for addressing most of my concerns and conducting additional experiments—I appreciate your effort. Please include them in a revision. However, I still disagree with using an image-trained VAE when audio-trained options are available, so I’m inclined to maintain my original rating.

---

> ### Author Response · Authors · 2024-11-25
> **Response to Reviewer mmH9's Feedback**
>
> Dear Reviewer mmH9,
>
> Thank you for your thoughtful feedback. We are currently exploring audio-specific VAEs, but due to the required extensive hyperparameter search and training, we were unable to incorporate these results within the discussion timeline. However, we believe that our proposed lightweight framework, featuring the Reducer and TAM components, offers significant benefits compared to the baseline VAE used in prior work. We appreciate your positive evaluation and will strive to include results utilizing an audio VAE once we have identified optimal hyperparameter configurations. We will incorporate the additional experiments recommended by the reviewers into the revised pdf.
>
> Best regards,
>
> The Authors

---

> > ### Author Response · Authors · 2024-11-29
> > **Update on AudioVAE Integration and Comparative Analysis (1/N)**
> >
> > Dear Reviewer mmH9,
> >
> > As previously mentioned, we have completed additional experiments to address your concern regarding the performance of our method with an audio-specific VAE. In these experiments, we used the AudioLDM VAE and trained our models with the same settings as for the image VAE, making only the necessary adjustments to input dimensions and parameters to accommodate the 1-channel audio VAE. This setup eliminates the channel selection issue that was present when using the image VAE.
> >
> > We find that while our method with the image VAE achieves state-of-the-art FAD and Alignment Accuracy, incorporating the AudioLDM VAE results in further improvements in audio quality. We have made demo samples using the AudioLDM VAE (with three models Tiny, Small, and Base) available via an anonymous link: [Google Drive](https://drive.google.com/file/d/1lNbjHuPuvDDUkhogSS37LXaU3bBrM9z-/view?usp=sharing), and we invite you to review them.
> >
> > In the revised PDF, we have clarified that our method supports both image and audio VAEs. Additionally, we have noted that channel selection is optional when combining MDSGen with the image VAE.
> >
> > Overall, the results remain consistent, with the AudioLDM VAE experiments reinforcing the advantages of MDSGen in key metrics, including state-of-the-art FAD and alignment accuracy, while significantly improving efficiency. Our smallest model, MDSGen-T (5M), achieves a FAD of 2.21, and the larger MDSGen-B (131M) improves it further to 1.34, setting a new benchmark.
> >
> > Below, we present side-by-side comparison tables for the Audio VAE and Image VAE, all models were trained with 300k steps:
> >
> > **Table 1. Performance Comparison: Image VAE vs. AudioLDM VAE**
> >
> > |       **Model**      |    **VAE**   |  **FID$\downarrow$**  |   **IS$\uparrow$**  |  **KL$\downarrow$**  | **Align. Acc$\uparrow$** |  **FAD$\downarrow$** |
> > |:--------------------:|:------------:|:---------:|:---------:|:--------:|:--------------:|:--------:|
> > |  **MDSGen-B (131M)** |   Image VAE  |   12.15   | **46.41** |   6.31   |    **98.58**   |    2.36   |
> > |                      | AudioLDM VAE | **11.83** |   30.37   | **6.03** |      96.52     | **1.34** |
> > |  **MDSGen-S (33M)**  |   Image VAE  | **12.88** | **40.39** |   6.21   |    **98.25**   |    2.75   |
> > |                      | AudioLDM VAE |   13.14   |   26.05   | **5.83** |      95.63     | **1.66** |
> > | **MDSGen-Tiny (5M)** |   Image VAE  | **14.78** | **33.23** |   6.23   |    **97.62**   |    3.69   |
> > |                      | AudioLDM VAE |   15.48   |   21.99   | **5.58** |      95.53     | **2.21** |
> >
> > **Table 2. Efficiency Comparison: Image VAE vs. AudioLDM VAE**
> >
> > |      **VAE**     | **Mel-spec resolution** | **Latent** | **#Tokens** | **Models** | **#params (M)** | **Infer time (s)$\downarrow$** | **#memory (M)$\downarrow$** |
> > |:----------------:|:------------:|:----------:|:-----------:|:----------:|-----------------|:------------------:|:---------------:|
> > |   **Image VAE**  |    128x512   |    16x64   |     256     |  MDSGen-T  |        5M       |      **0.01**      |     **1406**    |
> > |                  |              |            |             |  MDSGen-S  |       33M       |      **0.02**      |     **1508**    |
> > |                  |              |            |             |  MDSGen-B  |       131M      |      **0.05**      |     **2132**    |
> > | **AudioLDM VAE** |    64x816    |   16x204   |     816     |  MDSGen-T  |        5M       |        0.058        |       3087       |
> > |                  |              |            |             |  MDSGen-S  |       33M       |        0.114        |       5383      |
> > |                  |              |            |             |  MDSGen-B  |       131M      |        0.243        |       10165      |

---

> > > ### Author Response · Authors · 2024-11-29
> > > **Update on AudioVAE Integration and Comparative Analysis (2/N)**
> > >
> > > [continue]
> > >
> > > **Table 3. Masking Method using AudioLDM VAE**
> > >
> > > | **Masking** |  **FID$\downarrow$**  |   **IS$\uparrow$**  |  **KL$\downarrow$**  | **Align Acc$\uparrow$** |  **FAD$\downarrow$** |
> > > |:-----------:|:---------:|:---------:|:--------:|:-------------:|:--------:|
> > > |     SAM     |   11.90   |   28.54   | **5.83** |     96.25     |   1.52   |
> > > |     TAM     | **11.83** | **30.37** |   6.03   |   **96.52**   | **1.34** |
> > >
> > > Compared to existing video-to-audio methods, both VAEs show that MDSGen achieves state-of-the-art performance in FAD (1.34–2.16) and Alignment Accuracy (96.5%–98.5%). Note that, we reported FAD 2.16 of our MDSGen-B in the previous response from checkpoint 500k training steps with image VAE. While the Image VAE outperforms the AudioLDM VAE by about 2% in Alignment Accuracy, it falls behind in FAD (2.16 vs. 1.34), indicating that AudioLDM produces higher-quality audio. This discrepancy in alignment accuracy and IS may be attributed to several factors:
> > >
> > > 1. The alignment classifier is trained on 128×512 spectrograms, whereas AudioLDM uses 64×816 mel-spectrograms, inherently favoring features from the Image VAE, which aligns more closely with its input format.
> > > 2. Domain-Specific Design: AudioLDM is optimized for audio reconstruction, resulting in features less visually aligned with the mel-spectrograms expected by the classifier.
> > > 3. Resolution Differences: Image VAEs operate at higher spatial resolutions (128×512) and preserve finer, visually coherent patterns, leading to higher IS scores, while AudioLDM’s lower resolution (64×816) focuses on acoustic fidelity rather than visual consistency.
> > >
> > > Overall, we believe that the Image VAE performs better in IS and alignment accuracy due to its compatibility with visual-based metrics and the image-like nature of spectrograms. In contrast, Audio VAEs prioritize audio quality, creating a trade-off between perceptual fidelity and visual alignment in video-to-audio tasks.
> > >
> > > We hope these updated results address your concerns and look forward to further discussion.
> > >
> > > Best regards,
> > >
> > > Authors

---

### Official Review · Reviewer_Hij3 · 2024-11-03

**Soundness:** 3
**Presentation:** 4
**Contribution:** 3
**Rating:** 6
**Confidence:** 4

**Summary:**

The paper introduces MDSGen, an innovative framework designed for vision-guided open-domain sound generation with a focus on optimizing model parameter size, memory consumption, and inference speed. It features two major innovations: a redundant video feature removal reducer and a temporal-aware masking strategy. Utilizing denoising masked diffusion transformers, MDSGen achieves efficient sound generation without the need for pre-trained diffusion models. On the VGGSound dataset, the smallest MDSGen model demonstrates a 97.9% alignment accuracy while using fewer parameters, consuming less memory, and performing faster inference compared to the current state-of-the-art models. The results underscore the scalability and effectiveness of this approach.

**Strengths:**

1. The idea of compressing visual representations into one single vector is bold and intriguing, which reduces significant computing pressure on the DiT side.
2. The proposed TAM strategy is interesting and makes sense. Previous works about masking audio representations, such as AudioMAE, have drawn conclusions that the unstructured masking strategy is superior, which contradicts the conclusion in this paper. I believe this paper brings more insights into this topic.
3. The authors have conducted tons of ablation experiments to support their model design and parameter decision, making the conclusions plausible.

**Weaknesses:**

1. The major concern is about the modeling of audio representations, as I am familiar with this field. I believe that Mel spectrum is more of a 1D feature rather than 2D, because the spectrum does not satisfy translational invariance (if a formant chunk in a spectrum is moved from the bottom left to the top right, the semantics of the sound are likely to be completely destroyed), and the frequency domain and time domain cannot simply be simulated by spatial coordinates. A relevant observation in this paper is that a complete random masking strategy is underperformed by the temporal-aware masking strategy. Therefore, I believe that considering Mel spectrograms as gray-scale images and modeling them using 2D VAE pretrained with real images is suboptimal, which further prevents modeling sounds with varying lengths. There are already approaches that model audio using 1D VAE, such as Make-an-audio 2. So can the authors provide justifications for choosing 2D rather than 1D? In my view, choosing 1D combined with the TAM strategy could form a more compelling motivation.
2. The idea of compressing visual representations into one single vector is intriguing. However, I don't understand why this could work. How does one single vector provide accurate information about temporal position? I believe Diff-foley works because it adopts a sequence-to-sequence cross-attention mechanism, which provides rough sequential and positional information for the audio to follow. Could the author provide further analysis and discussion on this point? For example, analyzing the components related to temporal position within that vector, or the relation of the learned weights of reducer between key frames of videos.
3. Similar concern: the learned weights of reducer seem to be focused more on the head and tail frames of videos. Does this imply that the reducer is more focused on global video information? How can it be determined that it is capable of extracting local positional information?
4. The alignment classifier proposed in Diff-foley only reaches 90% accuracy on their test set. However, the best performance in this paper reaches 98+. How could this happen? Is the classifier involved during the training process?

**Questions:**

1. The reducer is designed to have fixed weights, serving as a weighted average of all the frames. However, the sound events of different videos are distinct, so why not adopt a dynamic weighted average strategy, e.g., attention pooling?
2. What is the exact implementation of the model? How is the visual conditioning (i.e., $p(x|v)$) implemented?

**Details Of Ethics Concerns:**

No concern.

---

> ### Author Response · Authors · 2024-11-20
> **AuthorResponse 1/N**
>
> Dear reviewer Hij3, thank you for recognizing our work and the helpful reviews, we would like to address your concerns point-by-point as below.
>
> > **Weakness 1.** The major concern is about the modeling of audio representations, as I am familiar with this field. I believe that Mel spectrum is more of a 1D feature rather than 2D, because the spectrum does not satisfy translational invariance (if a formant chunk in a spectrum is moved from the bottom left to the top right, the semantics of the sound are likely to be completely destroyed), and the frequency domain and time domain cannot simply be simulated by spatial coordinates. A relevant observation in this paper is that a complete random masking strategy is underperformed by the temporal-aware masking strategy. Therefore, I believe that considering Mel spectrograms as gray-scale images and modeling them using 2D VAE pretrained with real images is suboptimal, which further prevents modeling sounds with varying lengths. There are already approaches that model audio using 1D VAE, such as Make-an-audio 2. So can the authors provide justifications for choosing 2D rather than 1D? In my view, choosing 1D combined with the TAM strategy could form a more compelling motivation.
>
> **Re:**
> We appreciate the reviewer’s suggestion regarding the potential benefits of using a 1D VAE.
> Our choice to use a 2D VAE follows the prior work Diff-Foley [1] (NeurIPS 2023), a state-of-the-art on the task of video-to-audio with video-audio alignment. Diff-foley represents audio through 2D mel-spectrograms to achieve CAVP-aligned audio-video features in the first stage. For the diffusion process in the second stage, we also maintain this 2D mel-spectrogram format, as it allows straightforward replication to RGB-style inputs, making it compatible with pretrained image VAEs, which can efficiently encode the latent required for diffusion. The Diff-Foley analysis (Table 6 and Fig. 10) further supports this choice, showing that their use of an image-based VAE provides adequate reconstruction quality for audio.
>
> Additionally, using an image VAE allows us to leverage powerful image diffusion frameworks MDT [2] or Stable Diffusion. For example, Diff-Foley (NeurIPS 2023) demonstrates significant performance gains in the video-to-audio task by building on a pretrained image Stable Diffusion backbone.
>
> Given this context and to maintain a fair comparison, we adopted the same CAVP and image VAE components as Diff-Foley, while focusing on a more efficient framework with masked diffusion transformer instead of their heavy Unet Stable Diffusion.
>
> We have added this as a potential future direction in the revised manuscript’s limitations section.
>
> [1] Diff-Foley: Synchronized Video-to-Audio Synthesis with Latent Diffusion Models, NeurIPS 2023.
> [2] MDT: Masked Diffusion Transformer is a Strong Image Synthesizer, ICCV 2023
>
> > **Weakness 2.** The idea of compressing visual representations into one single vector is intriguing. However, I don't understand why this could work. How does one single vector provide accurate information about temporal position? I believe Diff-foley works because it adopts a sequence-to-sequence cross-attention mechanism, which provides rough sequential and positional information for the audio to follow. Could the author provide further analysis and discussion on this point? For example, analyzing the components related to temporal position within that vector, or the relation of the learned weights of reducer between key frames of videos.
>
> **Re:** After observing that existing video features contain significant redundancy, we designed the Reducer to eliminate this redundancy while retaining essential temporal information, easing the burden for the DiT model. As illustrated in the below figure, the reducer allocates temporal information across the 768-dimensional space. We draw the Figure of the Reducer design in the following link:
>
>  https://i.postimg.cc/Sx0VyxxZ/reducer.png
>
>
> The design works as follows: Input 32x512 → Linear → 32x768 → Conv1d → Output 1x768. As shown in the above figure, after the initial fully connected layer, each component (position) of the output vector $u_1$ contains the whole vector $v_1$ and retains local temporal information from the video.
> In the next layer, each component of the final 1x768 vector $v\in \mathbb{R}^{1\times 768}$ captures both local and global features from the original 32x512 video information, ensuring a comprehensive representation of the audio generation process. This lightweight vector significantly reduces the computational burden on DiT while preserving temporal information. This approach, where the Reducer distributes temporal information across the 768 dimensions, is explained in lines 189-193 of the paper.

---

> ### Author Response · Authors · 2024-11-20
> **AuthorResponse 2/N**
>
> > **Weakness 3.** Similar concern: the learned weights of reducer seem to be focused more on the head and tail frames of videos. Does this imply that the reducer is more focused on global video information? How can it be determined that it is capable of extracting local positional information?
>
> **Re:** As illustrated in the figure above, the second layer of the reducer is a 1x1 convolution that assigns weights to the 32 channels of the evolved video features. These features already encapsulate temporal information (local positional details) within each dimension of the 768-D output from the first layer, a linear transformation. We align with the reviewer’s hypothesis that the heavier weighting of certain head and tail elements indicates the reducer’s effort to extract global video information effectively.
>
> > **Weakness 4.** The alignment classifier proposed in Diff-foley only reaches 90% accuracy on their test set. However, the best performance in this paper reaches 98+. How could this happen? Is the classifier involved during the training process?
>
> **Re:** The classifier used for alignment evaluation is trained separately from our diffusion model; we simply use the pretrained classifier publicly provided in the Diff-Foley GitHub only for evaluation, it does not involve our training process.
> To clarify why alignment accuracy with **real audio and real video** pairs on the test set (90%) might be lower than with **generated audio and real video** pairs, we provide explanations of how is this possible as the following:
> 1. Firstly, real-life videos often contain various types of sound, a mixture of audio sources, including background music, noise, voice, and speech, which may vary significantly from the training data used to train the alignment classifier. This variability can lower alignment accuracy for real audio-video pairs.
> 2. In contrast, the generated audio is specifically conditioned on video content, aiming to produce sounds directly relevant to the video based on patterns that have been learned in training. As a result, the generated audio aligns more closely with the video content, leading to a higher audio-video alignment accuracy in the generated audio-video pairs.
> 3. For example, in the supplementary materials, the Ground Truth audio for the file named "-miI_C3At4Y_000104" contains strong background music that is unrelated to the video content, where a man is hitting badminton shots. In contrast, the audio generated by **Ours_Vocoder** for the same sample focuses only on the sound of a person playing badminton. We kindly invite the reviewer to verify this.
>
> Finally, the previous work Diff-Foley study reported alignment accuracies of their generation of about 94% (higher than their test set of 90%), using the same pre-trained classifier. Our results, reaching 98% alignment accuracy, are consistent with their findings, confirming that our model’s performance is reliable and not an outlier. Our model and code will be made publicly available for further research.
>
> > **Q1**. The reducer is designed to have fixed weights, serving as a weighted average of all the frames. However, the sound events of different videos are distinct, so why not adopt a dynamic weighted average strategy, e.g., attention pooling?
>
> **Re:** We conducted an experiment using attention pooling, as suggested by the reviewer, with results shown in the table below. While attention pooling delivers strong performance, it is slightly outperformed by the learnable weight approach. We hypothesize that the first layer of the Reducer extracts local information, while the second layer learns to assign weights to each frame's features, with a focus on key evolved frame features (heads/tails), enabling the extraction of more comprehensive global information.
>
> **Table 1. Choice of Pool Design for Reducer**
> |       Pooling method       | FID $\downarrow$ |     IS $\uparrow$    |    KL $\downarrow$   | Align. Acc. $\uparrow$ |
> |:--------------------------:|:----------------:|:---------:|:--------:|:-----------:|
> |    Naive Average Pooling   |       12.55      |   39.41   |   6.17   |    0.9848   |
> |      Attention Pooling     |       12.27      |   39.06   | **6.08** |    0.9847   |
> | Learnable Weight (default) |     **12.15**    | **46.33** |   6.30   |  **0.9858** |
>
> These results with attention pooling further demonstrate that a single vector is sufficient for audio generation, a finding that our work is the first to successfully apply to the video-to-audio task. We will make our code publicly available to allow the community to explore the effectiveness of this design and have included an analysis of different pooling methods in the revised paper.

---

> > ### Author Response · Authors · 2024-11-20
> > **AuthorResponse 3/N**
> >
> > > **Q2**. What is the exact implementation of the model? How is the visual conditioning (i.e., p(x|v)) implemented?
> >
> > **Re:** Exact implementation is that we processed video features dimention 32x512 to obtain a vector 768-D using Reducer with two layers (Linear + Conv1d) and then put lightweight vector into DiT framework via AdativeLayerNorm modulation (default code of DiT). Implementation of code for Reducer and p(v|x) are as below:
> >
> > **a) The pytorch code for Reducer module:**
> >
> >     # layer1
> >     self.mlp_feat = nn.Sequential(
> >             nn.Linear(in_features=512, out_features=768, bias=True),
> >             nn.GELU(),
> >             nn.LayerNorm(768, eps=1e-12)
> >             )
> >     # layer2
> >     self.conv1d_feat = nn.Sequential(
> >             nn.Conv1d(in_channels=32, out_channels=1, kernel_size=1),
> >             nn.GELU(),
> >             nn.LayerNorm(768, eps=1e-12)
> >             )
> >     # Usage of Reducer (layer1, layer2)
> >     v = self.mlp_feat(video_feat) # output dim 32x768, video_feat 32x512
> >     v = self.conv1d_feat(v)       # output dim 1x768
> >
> > **b) The pytorch code of p(x|v) with AdalayerNorm modulation (default DiT code):**
> >
> >     # hidden_size = 768
> >     def modulate(x, shift, scale):
> >         return x * (1 + scale.unsqueeze(1)) + shift.unsqueeze(1)
> >     self.adaLN_modulation = nn.Sequential(
> >             nn.SiLU(),
> >             nn.Linear(hidden_size, 6 * hidden_size, bias=True)
> >         )
> >     def forward(self, x, v): # x is mel-spec latent, v is condition vector
> >         shift_msa, scale_msa, gate_msa, shift_mlp, scale_mlp, gate_mlp = self.adaLN_modulation(v).chunk(6, dim=1)
> >         x = x + gate_msa.unsqueeze(1) * self.attn(modulate(self.norm1(x), shift_msa, scale_msa))
> >         return x

---

> > > ### Author Response · Authors · 2024-11-29
> > > **AuthorResponse 4/N**
> > >
> > > Dear Reviewer Hij3,
> > >
> > > Thank you for your insightful review. In addition to our previous response, we would like to share our updated results, where we integrate the AudioLDM VAE into our framework. While our method with the image VAE achieves state-of-the-art FAD and Alignment Accuracy, incorporating an audio-specific VAE further improves audio quality. We have made demo samples available through the anonymous link: [Google Drive](https://drive.google.com/file/d/1lNbjHuPuvDDUkhogSS37LXaU3bBrM9z-/view?usp=sharing).
> > >
> > > For a more detailed comparison of the quantitative metrics between the image VAE and AudioLDM VAE, kindly refer to our latest response to Reviewer #1 (GqkQ) at the top of this page.
> > >
> > > We appreciate your consideration of our previous response and the updated results, and we look forward to your feedback.
> > >
> > > Best regards,
> > >
> > > Authors

---

> ### Comment · Reviewer_Hij3 · 2024-12-03
> **Official Comment by Reviewer Hij3**
>
> Thank you for trying to address my concerns and providing additional results. I acknowledge the effort of incorporating a powerful masked diffusion transformer into the video-to-audio generation task, and the attempt to reduce the video features into a single vector. I also appreciate the clear illustration of the algorithm.
>
> Regarding the VAE issue, which other reviewers are also very concerned about, I don't think using an image-specific VAE is a reason to reject the paper. However, the authors should have recognized the drawbacks of the image-specific VAE when they discovered that the effects of temporal masking were better than purely random ones. This might also indicate that the Mel-spectrogram channel selection algorithm no longer represents an innovative aspect.
>
> In conclusion, I believe that this paper presents some interesting findings.

---

> ### Author Response · Authors · 2024-12-03
>
> **Dear Reviewer Hij3,**
>
> Thank you for your thoughtful feedback and for acknowledging our efforts to address your concerns. We appreciate your recognition of the integration of a masked diffusion transformer into the video-to-audio generation task, as well as your positive comments on the algorithm's clarity and our approach to video feature reduction.
>
> Regarding the VAE issue, we value your perspective and understand the importance of addressing the limitations of using an image-specific VAE. In our revision, we will include a detailed comparative analysis of both VAEs—highlighting their respective drawbacks and how they impact temporal masking effectiveness, channel selection, and overall performance. We agree that this additional analysis will provide a more comprehensive view of our method and clarify the trade-offs involved.
>
> Once again, thank you for your constructive feedback and for recognizing the contributions of our work.
>
> ---
> Best regards,
>
> **Submission265 Authors**

---

### Official Review · Reviewer_mkFe · 2024-11-03

**Soundness:** 2
**Presentation:** 2
**Contribution:** 2
**Rating:** 6
**Confidence:** 4

**Summary:**

The paper presents MDSGen, an efficient framework for vision-guided sound generation that minimizes model size, memory usage, and inference time. Key innovations include a temporal-aware masking strategy to enhance alignment accuracy and a redundant feature removal module to filter unnecessary video information. Using a lightweight masked diffusion transformer, MDSGen outperforms larger Unet-based models on VGGSound and Flickr-SoundNet, achieving high synchronization and alignment with significantly reduced computational costs.

**Strengths:**

The paper introduces a novel framework for video-to-audio sound generation that effectively combines a temporal-aware masking strategy with a redundant feature removal module.

MDSGen demonstrates significant improvements in model efficiency by using a smaller masked diffusion transformer architecture. The framework achieves high alignment accuracy on benchmark datasets with a fraction of the parameters, memory usage, and inference time compared to baselines.

The paper provides a structured explanation of MDSGen’s architecture and mechanisms, including the Temporal-Awareness Masking (TAM) and the Reducer module for filtering out redundant features. Extensive experimental results on VGGSound and Flickr-SoundNet datasets clearly validate the method’s effectiveness, with MDSGen achieving superior performance across alignment accuracy and efficiency metrics.

**Weaknesses:**

1. Lack of Novelty and Contribution: The paper presents the primary contributions are the Temporal-Awareness Masking (TAM) strategy and the visual Reducer module. However, masking strategies have been widely explored in audio generation research, as seen in works like [1, 2], and the specific concept of Temporal-Awareness Masking has been studied in [3, 4]. The visual Reducer module, primarily a 1x1 convolutional layer (line 181), lacks detailed design innovations, which limits its distinctiveness and impact.

2. Insufficient Exploration of Design Choices: For video-to-audio generation, the choice of video encoder plays a crucial role in understanding the video content. Clarification on the selection of CAVP as the video encoder would add valuable insight. Additionally, the paper could explore using more video encoders, such as CLIP [5], VideoMAE [6], ViVit [7], and TAM [8], which could enrich the technical depth of the proposed method.

3. Presentation and Writing:
Some claims in the paper lack supporting evidence, such as the statements in lines 183-185 that the proposed method “minimizes redundant features that could lead to overfitting” and in line 224 that setting N_2 = 4 “gives better performance for audio data.” These points would benefit from empirical support to substantiate their validity.

4. Supplementary Material: The quality of generated audio samples in the supplementary material raises concerns regarding the overall quality of results produced by the proposed method, which may affect its effectiveness and appeal.

[1]Pascual S, Yeh C, Tsiamas I, et al. Masked Generative Video-to-Audio Transformers with Enhanced Synchronicity[J]. arXiv preprint arXiv:2407.10387, 2024.

[2]Borsos Z, Sharifi M, Vincent D, et al. Soundstorm: Efficient parallel audio generation[J]. arXiv preprint arXiv:2305.09636, 2023.

[3]Bai H, Zheng R, Chen J, et al. A $^ 3$ T: Alignment-Aware Acoustic and Text Pretraining for Speech Synthesis and Editing[C]//International Conference on Machine Learning. PMLR, 2022: 1399-1411.

[4]Garcia H F, Seetharaman P, Kumar R, et al. Vampnet: Music generation via masked acoustic token modeling[J]. arXiv preprint arXiv:2307.04686, 2023.

[5]Radford A, Kim J W, Hallacy C, et al. Learning transferable visual models from natural language supervision[C]//International conference on machine learning. PMLR, 2021: 8748-8763.

[6]Tong Z, Song Y, Wang J, et al. Videomae: Masked autoencoders are data-efficient learners for self-supervised video pre-training[J]. Advances in neural information processing systems, 2022, 35: 10078-10093.

[7]Arnab A, Dehghani M, Heigold G, et al. Vivit: A video vision transformer[C]//Proceedings of the IEEE/CVF international conference on computer vision. 2021: 6836-6846.

**Questions:**

Please see Weaknesses.

---

> ### Author Response · Authors · 2024-11-20
> **AuthorResponse 1/N**
>
> Dear reviewer mkFe, thank you for your insightful comments, we would like to address your concerns one by one as follows.
>
> > **Weakness 1.** Lack of Novelty and Contribution: The paper presents the primary contributions are the Temporal-Awareness Masking (TAM) strategy and the visual Reducer module. However, masking strategies have been widely explored in audio generation research, as seen in works like [1, 2], and the specific concept of Temporal-Awareness Masking has been studied in [3, 4]. The visual Reducer module, primarily a 1x1 convolutional layer (line 181), lacks detailed design innovations, which limits its distinctiveness and impact.
>
> **Re:** We would like to elaborate on your concern in two parts:
>
> **1.1. Novelty on masking branch:** When reviewing the literature on video-to-audio tasks, to the best of our knowledge, our work is the first to leverage successfully the masked diffusion transformers MDT [P1] (ICCV 2023), the state-of-the-art framework in ImageNet generation, expanding its advantage on this task. It is different from the four works mentioned by the reviewer in several aspects as follows:
>
> 1. We would like to clarify that the works referenced by the reviewer in [1,2] are based on fundamentally different frameworks from our approach. Specifically, [1,2] utilize a technique (non-diffusion) that masks discrete token indices and then predicts these masked indices within a codebook using cross-entropy loss, with the generation objective with **mask used in both training and inference (i.e., generating discrete indices)**. In contrast, our approach is diffusion-based, where the objective is to denoise continuous latent. The concept of masking diffusion models was introduced recently in the MDT paper [P1], where it functions as a regularizer for the transformer but it is **used only in training, and the mask branch is removed during inference**, differing from the mask generative methods in [1,2].
> 2. Regarding [3,4], while these works also involve temporal masking, key differences set our approach apart. **First**, [3] apply masking directly to the pixel space of mel-spectrograms, whereas we implement masking in the latent space of mel-spectrograms. **Second**, [3,4] mask clean mel-spectrograms (i.e., noise-free), whereas our approach masks noisy mel-spectrogram latents (Gaussian noises), adding robustness within the diffusion framework. **Third**, [3,4] explore masking in the non-diffusion framework with the task of **text-to-audio and speech editing**, where we are the first to explore masked diffusion transformer MDT [P1] on the task of **video-to-audio**.
>
> [P1] MDT: Masked Diffusion Transformer is a Strong Image Synthesizer, ICCV 2023
>
> **1.2. Contribution of our Reducer and its detail**
> To the best of our knowledge, our work is the first to demonstrate that a single vector representation of a video can sufficiently capture the information needed for video-to-audio generation. When combined with an ultra-lightweight model (5M parameters), our approach outperforms an existing 860M model in terms of audio-video alignment scores. This finding may offer valuable new insights for the field.
>
> Specifically, our Reducer included two layers: Linear(512, 768) and Conv1d(32, 1). Given the video features V of shape (32, 512), the first layer of Reducer projects it to obtain the feature (32, 768), and the second layer further transforms that feature to lightweight vector $v$ (1x768) to the DiT backbone. We provide details of our Reducer both schematic and code as follows:
> 1. A figure to illustrate our Reducer is available at this link: https://i.postimg.cc/Sx0VyxxZ/reducer.png
> 2. Code pytorch implementation:
>
> Pytorch pseudo code of Reducer with (layer1, layer2):
>
>     class Reducer(self):
>         # layer1
>         self.mlp_feat = nn.Sequential(
>                 nn.Linear(in_features=512, out_features=768, bias=True),
>                 nn.GELU(),
>                 nn.LayerNorm(768, eps=1e-12)
>                 )
>         # layer2
>         self.conv1d_feat = nn.Sequential(
>                 nn.Conv1d(in_channels=32, out_channels=1, kernel_size=1),
>                 nn.GELU(),
>                 nn.LayerNorm(768, eps=1e-12)
>                 )
>     # Usage of Reducer (layer1, layer2)
>     v = self.mlp_feat(video_feat) # output dim 32x768, video_feat 32x512
>     v = self.conv1d_feat(v)       # output dim 1x768
>
> We have revised our paper to include comparisons with these related works, and details of our Reducer, as red color in the updated sections.

---

> ### Author Response · Authors · 2024-11-20
> **AuthorResponse 2/N**
>
> > **Weakness 2.** Insufficient Exploration of Design Choices: For video-to-audio generation, the choice of video encoder plays a crucial role in understanding the video content. Clarification on the selection of CAVP as the video encoder would add valuable insight. Additionally, the paper could explore using more video encoders, such as CLIP [5], VideoMAE [6], ViVit [7], and TAM [8], which could enrich the technical depth of the proposed method.
>
> **Re:** We selected the CAVP video encoder based on the findings of prior work, Diff-Foley [P2], which conducted an extensive ablation study comparing various video encoders, including CLIP and ResNet50. Their results demonstrated that CAVP outperformed these alternatives in both FID and audio-video alignment accuracy (kindly refer to Table 2 in [P2]), two metrics critical for video-to-audio generation, where precise alignment is essential.
>
> Exploring the potential of other video encoders, such as VideoMAE and ViViT, is an exciting avenue that we plan to pursue in future work.
>
> [P2] Diff-Foley: Synchronized Video-to-Audio Synthesis with Latent Diffusion Models, NeurIPS 2023.
>
> > **Weakness 3.** Presentation and Writing: Some claims in the paper lack supporting evidence, such as the statements in lines 183-185 that the proposed method “minimizes redundant features that could lead to overfitting” and in line 224 that setting N_2 = 4 “gives better performance for audio data.” These points would benefit from empirical support to substantiate their validity.
>
> **Re:** Thanks for your concern, we have provided the experimental results below figure, and have included them in our revised paper Appendix.
>
> **Overfitting phenomenon with redundant features.** Minimizing redundant features helps prevent overfitting as shown in this Figure: https://i.postimg.cc/3NY4Lz8F/overfit-features.png
>
> **Ablation with the number of decoder layers**. We provide the ablation results in below table:
>
> **Table 1. The position of $N_2=2, N_2=4$ decoder layers.**
> |      Decoder      | FID$\downarrow$ | IS $\uparrow$ | KL $\downarrow$ | Align. Acc.$\uparrow$ |
> |:-----------------:|:---------------:|:-------------:|:---------------:|:---------------------:|
> |      $N_2=2$      |      12.46      |   **47.99**   |       6.43      |         0.9823        |
> | $N_2=4$ (default) |    **12.15**    |     46.33     |     **6.30**    |       **0.9858**      |
> > **Weakness 4.** Supplementary Material: The quality of generated audio samples in the supplementary material raises concerns regarding the overall quality of results produced by the proposed method, which may affect its effectiveness and appeal.
>
> **Re:** We acknowledge that the waveform quality in the initial submitted supplementary files may be not ideal due to the limitations of the Griffin-Lim algorithm, which can introduce distortions during the mel-spectrogram-to-waveform conversion.
>
> To address this, we conducted a new experiment using the neural vocoder HiFi-GAN, trained on VGGSound, as a replacement for the baseline Griffin-Lim. This resulted in significant improvements in audio quality, as evidenced by the updated Fréchet Audio Distance (FAD) scores and subjective human evaluation results, detailed in the tables below.
>
> Updated audio samples have been included in the new supplementary materials, and we kindly invite you to review them for improved quality.
>
> **Table 2. FAD scores on waveform**
> | Method | See-and-hear (vocoder) | FoleyCrafter (vocoder) | Diff-Foley (Griffin-Lim) | Diff-Foley (vocoder) | MDSGen (Ours) (Griffin-Lim) | MDSGen (Ours) (vocoder) |
> |:------:|:----------------------:|:----------------------:|:------------------------:|:--------------------:|:---------------------------:|:-----------------------:|
> |   FAD $\downarrow$  |          5.55          |          2.45          |           6.08           |         4.71         |             4.37            |         **2.16**        |
>
> **Table 3. Human evaluation with a subjective listening test**
> | Metric\Method |  See-and-hear |  FoleyCrafter |   Diff-Foley  |  MDSGen-B (Ours)  |  Ground Truth  |
> |:-------------:|:-------------:|:-------------:|:-------------:|:-----------------:|:--------------:|
> |     MOS-AQ $\uparrow$   | 2.68$\pm$0.25 | 3.21$\pm$0.23 | 3.29$\pm$0.24 | **3.66$\pm$0.23** | 4.74 $\pm$0.12 |
> |     MOS-AV $\uparrow$    | 2.95$\pm$0.20 | 3.44$\pm$0.26 | 3.56$\pm$0.23 | **3.76$\pm$0.21** |  4.62$\pm$0.23 |
>
> Futher setup of the human evaluation, kindly refer to our common response.

---

> > ### Author Response · Authors · 2024-11-29
> > **AuthorResponse 3/N**
> >
> > Dear Reviewer mkFe,
> >
> > Thank you for your thoughtful review. In addition to our previous response, we invite you to explore our new results, where we incorporate the AudioLDM VAE into our framework. While our method with the image VAE achieves state-of-the-art FAD and Alignment Accuracy, using an audio-specific VAE further enhances audio quality. We have provided improved demo samples with AudioLDM VAE via the anonymous link: [Google Drive](https://drive.google.com/file/d/1lNbjHuPuvDDUkhogSS37LXaU3bBrM9z-/view?usp=sharing).
> >
> > We look forward to your feedback on our previous rebuttal and the newly updated results.
> >
> > Best regards,
> >
> > Authors

---

> ### Author Response · Authors · 2024-12-02
>
> Dear Reviewer mkFe,
>
> We hope our rebuttal and the additional materials address your concerns. We would appreciate any feedback or further concerns you may have. Thank you for your time and consideration.
>
> Best regards,
>
> Submission265 Authors

---

### Official Review · Reviewer_GqkQ · 2024-11-04

**Soundness:** 2
**Presentation:** 4
**Contribution:** 3
**Rating:** 6
**Confidence:** 4

**Summary:**

**Update after discussion period**:

My biggest concern of this paper is about its audio reconstruction pipeline. The original pipeline consists of an RGB VAE and a non-DNN mel2wave conversion module, resulting in terrible sound quality. Moreover, such a pipeline has been used in video2audio community widely, which is a pity.

During the rebuttal period, the authors not only added the experiment of using a vocoder for the mel2wave conversion, but also re-trained their models with an audio VAE. As a result of their efforts, the FAD score improved from 4.37 to 2.16 and eventually 1.34. Congrats on the achievement!

I listened to the latest samples in the Google Drive, and found a distinct advantage of audio VAE in samples that contain non-noise or musical sources (sample027__SrU3mfTPYg_000032.mp4, sample045_2es7oZzwLWM_000030.mp4). Although the AV-align score is slightly worse with an audio VAE, the score is still higher than conventional methods by a large margin.

Since the AV-align score is computed by a pretrained DNN model, moreover, considering the gap between v2a community and audio generation community, the metric might not be very reliable. For example, in samples such as "sample007_-2sOH8XovEE_000484.mp4", I don't feel the alignment is as good as the scores indicate. Perhaps some future works can be done to further improve the metric itself.

I increased my ratings to **encourage further collaboration between video2audio and audio communities**.

---------------

The paper proposes "MDSGen", an efficient model based on Masked Diffusion Transformer, for video-to-audio generation.

The challenges of video-to-audio generation are mainly:
1. Heavy computation and memory usage;
2. Requirements for the audio quality;
3. Requirements for the audio-video alignment;

MDSGen reduces the resource consumption by using very light-weight Transformer coupled with fast diffusion samplers such as DPM solver, as well as a dimension reduction module to reduce the size of the video conditioning embeddings.

MDSGen improves audio quality and audio-video quality by introducing a time-aware masking strategy into the mask DiT framework, together with other efforts.

Conceptualy, MDSGen looks like a framework that replaces the "text prompt" in text-to-audio DiT [StableAudioOpen],[MakeAnAudio2] by a video feature embedding. Hence the technical contributions are more in micro aspects.

However, some design choices may have severely affected the audio quality, making the work less solid or reusable to the community. Audio quality observed in the supplementary files is far from the level in modern text-to-audio models such as [AudioLDM], [MakeAnAudio2], [SpecMaskGIT], [StableAudioOpen].

**Strengths:**

Most contributions of MDSGen are about micro design aspects.
1. Channel selection of Mel-spec
2. Time-aware masking strategy for generative models
3. Reduced dimension of the video features
4. Small model size and fast inference speed

**Weaknesses:**

## Major issues
### 1. Improper audio reconstrution pipeline
MDSGen ustilizes the VAE from Stable Diffusion, which is not trained for Mel-spec. Although the authors carefully discussed how to take the most advantage of this image VAE, the discussion itself is **NOT** reusable for the audio community, as there have been plenty of audio VAE designs, some of which are publicly available such as [AudioLDM], [MakeAnAudio2], [StableAudioOpen], [DAC].

Another improper choice is that, MDSGen utilizes the Griffin-Lim Algorithm (GLA) to convert mel-spec back to wave forms. GLA has almost been abandoned by audio community, due to the recent advance in neural vocoder, e.g., [HiFiGAN], [UnivNet], [BigVGAN]. I believe the apparent phase distortion in the supplementary files might have been caused by GLA.

There is a rough comparison on the quality of audio reconstruction pipeline in a recent paper [SpecMaskGIT], I hope it could be useful for the improvement of MDSGen.

I strongly recommend the authors to consider audio-specified reconstruction pipelines for improved audio quality. Even in audio-visual generation community, we can see the usage of such audio VAE for excellent audio quality, e.g., [VisualEchoes]
### 2. Invalid claims on the result
Because the audio quality is far from the baseline in audio generation community, it is improper to claim that MDSGen is better in "audio-video alignment".

I believe, the audio-video alignment can be evaluated only when the audio quality is sufficiently good. Given the current audio quality, I don't think the model is ready for further evaluation.
## Minor issues
### 1. Evaluation metrics
The FID used in this paper comes from the implementation of SpecVQGAN, a pioneer of audio generation. However, the FAD implementation ([AudioLDM],[FAD_github]) has been more widely accepted in audio community. Evaluating with the widely adopted FAD metric can also help the readers to compre the audio quality with other audio generation models.
### 2. Insufficient ablation study
MDSGen trains a learnable module to reduce the video feature sequence into a single vector. From Figure 6, we can observe that the learned weights are quite evenly distributed (except the beginning and ending frames).

The observation posts a question: How much improvement can the learnable reducer bring compared to a naive average pooling?

[StableAudioOpen]: https://arxiv.org/abs/2407.14358
[MakeAnAudio2]: https://arxiv.org/abs/2305.18474
[AudioLDM]: https://audioldm.github.io/
[SpecMaskGIT]: https://arxiv.org/abs/2406.17672
[DAC]: https://github.com/descriptinc/descript-audio-codec
[HiFiGAN]: https://github.com/jik876/hifi-gan
[UnivNet]: https://github.com/rishikksh20/UnivNet-pytorch
[BigVGAN]: https://github.com/NVIDIA/BigVGAN
[VisualEchoes]: https://arxiv.org/abs/2405.14598
[AudioLDMEval]: https://github.com/haoheliu/audioldm_eval
[FAD_github]: https://github.com/gudgud96/frechet-audio-distance

**Questions:**

Three models are presented in the paper (Tiny, Small and Base) except the overfitting large model. What is the model presented in the supplementary files?

Is there any reason to use an image VAE instead of audio VAEs?

Did the authors observe any advantage of GLA over a neural vocoder?

Is it possible to run a subjective listening test, and see the consistency between human evaluation and the audio-video alignment accuracy measured by a DNN model?

Could the authors measure the FAD scores on top of the current FID scores?

---

> ### Author Response · Authors · 2024-11-20
> **AuthorResponse 1/N**
>
> Dear reviewer GqkQ, thank you for your concerns, we would like to explain your concerns point by point in three parts: **I. Major issues, II. Minor issues, and III. Questions.**
>
> **Part I. Major issues**
> > **Weakness 1.1.** Improper audio reconstrution pipeline
> > 1.1.1 MDSGen ustilizes the VAE from Stable Diffusion, which is not trained for Mel-spec. Although the authors carefully discussed how to take the most advantage of this image VAE, the discussion itself is NOT reusable for the audio community, as there have been plenty of audio VAE designs, some of which are publicly available such as AudioLDM, MakeAnAudio2, StableAudioOpen, DAC.
> > 1.1.2. There is a rough comparison on the quality of audio reconstruction pipeline in a recent paper SpecMaskGIT, I hope it could be useful for the improvement of MDSGen.
> > 1.1.3. I strongly recommend the authors to consider audio-specified reconstruction pipelines for improved audio quality. Even in audio-visual generation community, we can see the usage of such audio VAE for excellent audio quality, e.g., VisualEchoes
>
> **Re:** We chose the Stable Diffusion VAE for several key reasons:
> 1. **Follow prior works [1] and [3] to use the VAE.** Diff-Foley [1] (NeurIPS 2023) was the first in applying diffusion models to Video-to-audio task and proposed to use the Stable Diffusion VAE for mel-spectrograms in the video-to-audio task. **This VAE demonstrated strong reconstruction quality for mel-spectrograms** despite its training on images as **detailed in Section D (page 20)** of Diff-Foley paper, with their evaluations **in Fig. 10 and Tab. 6 on page 21** (we invite the reviewer to kindly have a double check). The recent study Action2Sound [3] (ECCV24) also employs the Stable Diffusion VAE for mel-spectrograms, further validating its effectiveness in this context. This reaffirms that Stable Diffusion’s VAE remains a viable and reusable option for audio mel-spectrogram reconstruction.
> 2. **Mel-spectrogram is represented in 2D (128x512), it is easy to get image by repeating channel dimension** so that it is easily adopted the power of existing diffusion frameworks such as Stable Diffusion or DiT [4] or MDT [2]. Therefore, without focusing to replicate the reconstruction experiments, we decided to use image VAE for mel-spectrogram following prior works [1,3] that used for the audio data.
> 3. **To ensure a fair comparison with our main baseline**, Diff-Foley [1], a state-of-the-art in the task of video-guided open-domain sound generation with video-audio alignmen score. The main focus of our paper is to propose to replace the heavy Stable Diffusion backbone of existing Diff-Foley, based on a new recent advanced technique namely masked diffusion transformers MDT [2] (ICCV23), a state-of-the-art on ImageNet. Our method is a much more efficient alternative to Diff-Foley (NeurIPS23) which rely on heavy Unet diffusion backbones. So we keep other things like CAVP and VAE the as same Diff-Foley as possible.
>
> Using the same VAE from Stable Diffusion as our baseline, our method outperforms existing approaches across multiple metrics (FAD, MOS, Align. Acc) while being significantly more efficient in terms of model size and computational speed. We anticipate that integrating a VAE from audio-specific models such as AudioLDM or Make-An-Audio2 could further enhance our framework.
>
> > **Weakness 1.2.** algorithms convert mel-spec back to wave forms
> > 1.2. Another improper choice is that, MDSGen utilizes the Griffin-Lim Algorithm (GLA) to convert mel-spec back to wave forms. GLA has almost been abandoned by audio community, due to the recent advance in neural vocoder, e.g., HiFiGAN, UnivNet, BigVGAN. I believe the apparent phase distortion in the supplementary files might have been caused by GLA.
>
> **Re:** The choice of conversion algorithms for transforming mel-spectrograms back into waveforms is orthogonal to our work. We fully agree that using a more advanced algorithm for this conversion would further enhance the overall framework. To ensure a fair comparison and emphasize the strengths of our approach relative to existing methods in the video-to-audio task, we followed the prior work Diff-Foley (NeurIPS 2023) and used the same Griffin-Lim algorithm provided in their published code.
>
> **In response to the reviewer’s suggestion, we implemented a neural vocoder to replace Griffin-Lim. This change resulted in an improvement in waveform quality, reflected in the FAD scores as shown in the next response.**
>
>
> [1] Diff-Foley: Synchronized Video-to-Audio Synthesis with Latent Diffusion Models, NeurIPS 2023.
>
> [2] MDT: Masked Diffusion Transformer is a Strong Image Synthesizer, ICCV 2023
>
> [3] Action2Sound: Ambient-Aware Generation of Action Sounds from Egocentric Videos, ECCV 2024.
>
> [4] Scalable Diffusion Models with Transformers, ICCV 2023.

---

> > ### Author Response · Authors · 2024-11-20
> > **AuthorResponse 2/N**
> >
> > > **Weakness 2.1.** Because the audio quality is far from the baseline in audio generation community, it is improper to claim that MDSGen is better in "audio-video alignment".
> >
> > **Re:** While audio generation models in text-to-audio tasks (e.g., AudioLDM, MakeAnAudio2, SpecMaskGIT, StableAudioOpen) have shown impressive results, video-to-audio generation remains a uniquely challenging task. It requires generating audio that aligns realistically with both the visual content and dynamic events in videos.
> >
> > The video-to-audio domain is still emerging and challenging, with Diff-Foley (NeurIPS 2023) being the first diffusion-based model for this task. Our claim that MDSGen improves audio-video alignment is purely based on the objective metric of alignment accuracy. This metric, introduced by Diff-Foley, employs a classifier to predict the alignment relevance of an audio-video pair.
> >
> > **As per the reviewer’s suggestion, now we have trained a HiFi-GAN vocoder to enhance the audio quality. This neural vocoder improved Fréchet Audio Distance (FAD) and human evaluation MOS (in the next response). The updated audio samples, generated by replacing Griffin-Lim with HiFi-GAN, can be accessed through the new supplementary materials. Note that, our model architecture and its generated mel-spectrogram remain unchanged, with only the vocoder swapped to improve sound quality.**
> >
> >
> > > **Weakness 2.2.** I believe, the audio-video alignment can be evaluated only when the audio quality is sufficiently good. Given the current audio quality, I don't think the model is ready for further evaluation.
> >
> > **Re:** Thanks for your great concern that helps us further improve our model performance with a newly trained vocoder. We find that the audio quality in the submitted supplementary, which indeed has some distortions, is due to the Griffin-Lim algorithm's limitation.
> > In response, we have replaced Griffin-Lim with a neural vocoder, HiFi-GAN, trained by ourselves, which has led to an improvement in audio quality and FAD score (in the next response). Updated audio samples generated with the new vocoder are available via the new supplementary. **Note that the mel-spectrograms produced by our model remain unchanged**, preserving the core functionality of our proposed method, as this adjustment only affects the mel-to-waveform conversion and the FAD score. All other metrics (FID, IS, KL, Algn. Acc.) in our paper are not impacted since they are evaluated directly using the generated mel-spectrogram.
> >
> > For open-domain sound generation from video inputs, we recognize that there is considerable potential for improvement. Given our current constraints, such as training on the VGGSound dataset, the audio quality may not yet fully reflect the model’s potential. However, in **comparing our work with existing baselines** in the task of video-to-audio such as Diff-Foley, See-And-Hear, and FoleyCrafter, our model outperformed them on multiple metrics while being much more faster and efficient with **very low cost**.
> >
> > We include results from human evaluations regarding the audio quality, which address your next question.

---

> ### Author Response · Authors · 2024-11-20
> **AuthorResponse 3/N**
>
> **Part II. Minor issues**
>
> > 1. Evaluation metrics
> The FID used in this paper comes from the implementation of SpecVQGAN, a pioneer of audio generation. However, the FAD implementation (AudioLDM,FAD_github) has been more widely accepted in audio community. Evaluating with the widely adopted FAD metric can also help the readers to compre the audio quality with other audio generation models.
>
> **Re:** We provide the evaluation on FAD as below. The results show that when using a neural vocoder, our model achieves state-of-the-art FAD on the VGGSound test set with an FAD of 2.16. With the simple Griffin-Lim, our MDSGen outperforms two out of three methods with an FAD of 4.37.
>
> **Table 1. FAD scores on waveform**
> | Method | See-and-hear (vocoder) | FoleyCrafter (vocoder) | Diff-Foley (Griffin-Lim) | Diff-Foley (vocoder) | MDSGen (Ours) (Griffin-Lim) | MDSGen (Ours) (vocoder) |
> |:------:|:----------------------:|:----------------------:|:------------------------:|:--------------------:|:---------------------------:|:-----------------------:|
> |   FAD $\downarrow$  |          5.55          |          2.45          |           6.08           |         4.71         |             4.37            |         **2.16**        |
>
>
> > 2. Insufficient ablation study
> MDSGen trains a learnable module to reduce the video feature sequence into a single vector. From Figure 6, we can observe that the learned weights are quite evenly distributed (except the beginning and ending frames).
> The observation posts a question: How much improvement can the learnable reducer bring compared to a naive average pooling?
>
> **Re:** Thanks for the suggestion, we conducted an experiment using the naive average pooling, the results are reported in below table:
>
> **Table 2. Choice of Pool Design for Reducer Layer 2**
> |       Pooling method       | FID $\downarrow$ |     IS$\uparrow$    |    KL $\downarrow$    | Align. Acc. $\uparrow$ |
> |:--------------------------:|:----------------:|:---------:|:--------:|:-----------:|
> |    Naive Average Pooling   |       12.55      |   39.41   |   6.17   |    0.9848   |
> |      Attention Pooling     |       12.27      |   39.06   | **6.08** |    0.9847   |
> | Learnable Weight (default) |     **12.15**    | **46.33** |   6.30   |  **0.9858** |
>
> We have added an analysis of these results in the revised pdf in Appendix Section A.3.
>
> **Part III. Questions**
> > **Q1.** Three models are presented in the paper (Tiny, Small and Base) except the overfitting large model. What is the model presented in the supplementary files?
>
> **Re:** It is an MDSGen-B model (131MB) with a Griffin-Lim algorithm. Now, we have added the vocoder version in the new supplementary for comparison, kindly check it.
>
> > **Q2.** Is there any reason to use an image VAE instead of audio VAEs?
>
> **Re:** Kindly refer to three key reasons for using image VAE in our response to the **weakness 1.1** part above.
>
> > **Q3.** Did the authors observe any advantage of GLA over a neural vocoder?
>
> **Re:** One advantage of using the Griffin-Lim Algorithm (GLA) is its simplicity; it does not require storing model parameters or pre-trained weights and does not depend on extensive training. In contrast, neural vocoders, require substantial training, which incurs additional computational costs and complexity. For example, we now have trained HifiGAN on VGGSound from scratch and its checkpoint is about 918M including parameters and optimizer, took 3 days on 2 A100 GPUs.
>
> Existing pretrained vocoder: we applied the public pretrained checkpoints of HifiGAN or AudioLDM into our mel-spectrogram but it didn't give satisfactory results, because the mel-spectrogram we used is different from the mel-spectrogram of those work (dimension, fmax, fmin, hoplen, etc...).
> Newly trained vocoder: Now, we have trained the vocoder HifiGAN from scratch on the VGGSound dataset and we find that the vocoder improved audio quality than the baseline Griffin algorithm. Note that, this improvement comes only from the change of vocoder, our generated model remains the same.

---

> > ### Author Response · Authors · 2024-11-20
> > **AuthorResponse 4/N**
> >
> > > **Q4.** Is it possible to run a subjective listening test, and see the consistency between human evaluation and the audio-video alignment accuracy measured by a DNN model?
> >
> > **Re:** We conducted a human evaluation by generating 50 audio samples based on 50 videos for each method. Five participants were asked to evaluate each method. Participants were instructed to watch the videos and listen to the corresponding audio, rating each on a scale from 1 to 5 based on the following criteria:
> > **1) Audio Quality (AQ):** How good is the sound quality?
> > **2) Audio-Video Content Alignment (AV):** How well does the sound match the video content?
> > The mean opinion scores (MOS) for each metric (ranging from 1 to 5) are reported in the table below:
> >
> > **Table 3. Human evaluation with a subjective listening test**
> > | Metric\Method |  See-and-hear |  FoleyCrafter |   Diff-Foley  |  MDSGen-B (Ours)  |  Ground Truth  |
> > |:-------------:|:-------------:|:-------------:|:-------------:|:-----------------:|:--------------:|
> > |     MOS-AQ $\uparrow$    | 2.68$\pm$0.25 | 3.21$\pm$0.23 | 3.29$\pm$0.24 | **3.66$\pm$0.23** | 4.74 $\pm$0.12 |
> > |     MOS-AV $\uparrow$    | 2.95$\pm$0.20 | 3.44$\pm$0.26 | 3.56$\pm$0.23 | **3.76$\pm$0.21** |  4.62$\pm$0.23 |
> >
> >
> > > **Q5.** Could the authors measure the FAD scores on top of the current FID scores?
> >
> > **Re:** We report the FAD score on top of FID scores as below table. Note that, Diff-Foley and MDSGen used the generated mel-spectrogram directly to calculate the FID metric so we keep it unchanged with the case of using a Vocoder and Griffin-Lim.
> >
> > **Table 4. FAD scores on top of FID scores**
> > | Metric\Method | See-and-hear (vocoder) | FoleyCrafter (vocoder) | Diff-Foley (Griffin-Lim) | Diff-Foley (vocoder) | MDSGen (Ours) (Griffin-Lim) | MDSGen (Ours) (vocoder) |
> > |:------:|:----------------------:|:----------------------:|:------------------------:|:--------------------:|:---------------------------:|:-----------------------:|
> > |   FAD $\downarrow$  |          5.55          |          2.45          |           6.08           |         4.71         |             4.37            |         **2.16**        |
> > |   FID $\downarrow$  |          21.35          |          12.07          |           **10.55**           |         -         |             11.19            |         -        |

---

> ### Author Response · Authors · 2024-12-03
>
> **Dear Reviewer GqkQ,**
>
> Thank you for your thoughtful and encouraging feedback during the rebuttal process. We sincerely appreciate your recognition of our efforts to address the concerns regarding the audio reconstruction pipeline, including the integration of a neural vocoder and the re-training with an audio VAE. Your acknowledgment of the improvements, particularly the significant enhancement in FAD scores and the qualitative assessment of the generated samples, means a lot to us.
>
> **We are especially grateful that you increased your score, moving our submission toward acceptance**. Your support and encouragement are highly motivating, and we value your insights on the importance of bridging the gap between the video-to-audio and audio generation communities.
>
> Additionally, your suggestion to explore more reliable alignment metrics is well taken, and we look forward to contributing to this area in future work.
>
> Thank you once again for your constructive feedback and for supporting our work.
>
> ---
> Best regards,
>
> **Submission265 Authors**

---

### Author Response · Authors · 2024-11-20
**Rebuttal Summary**

Dear reviewers **GqkQ, mkFe, Hij3** and **mmH9**

Thank you for your reviews and patience. We have prepared our rebuttal, providing detailed explanations and conducting the necessary experiments as per your suggestions. Each of your concerns has been addressed individually in the discussion panel:

1. We have **updated a revised PDF**, highlighting changes in red (both the main paper and Appendix).
2. We have included new demo samples generated using the **neural vocoder HiFi-GAN instead of the baseline Griffin-Lim algorithm** (we invite reviewers to kindly check it, the WAV is included in each video). The new vocoder shows improvement in sound quality and FAD scores compared to the previous samples.
3. We use [FAD_github](https://github.com/gudgud96/frechet-audio-distance) to **compute the FAD score** on waveforms. The results show that our model achieves state-of-the-art FAD metric when using a neural vocoder. Note that, our model results on other metrics (FID, IS, KL, Align. Acc.) remain unaffected, as they are directly evaluated based on the generated mel-spectrograms.
4. We conducted a **subjective human evaluation** using the mean opinion score (MOS) for a total of 250 videos, by generating 50 audio samples based on 50 test videos for each method. Five participants were asked to evaluate the results from each method, assigning scores from 1 to 5 based on the following criteria:
    + Audio Quality (AQ): How good is the sound quality?
    + Audio-Video Alignment (AV): Does the sound relate to the video content?

We invite you to review the detailed rebuttal in the discussion panel. Thank you.

---
Best regards,

Submission265 Authors

---

### Meta-Review · Area_Chair_YHSs · 2024-12-21

**Metareview:**

This paper introduces "MDSGen," an efficient model based on a Masked Diffusion Transformer, for video-to-audio generation. MDSGen addresses several key challenges in this area, including: (1) high computational and memory demands; (2) the need for high-quality audio output; and (3) the importance of audio-visual alignment. Four expert reviewers initially raised concerns regarding the audio reconstruction pipeline, the validity of certain claims about the results, and the choice of audio representation. The authors responded with extensive experiments and clarifications.

Following discussion, most major concerns were resolved. All reviewers ultimately rated the paper a 6 (marginally above the acceptance threshold). I concur with the reviewers' assessment and believe this paper, exploring both multimodal generative AI and efficient generation, is a valuable contribution to ICLR and an important topic in the field. The authors should incorporate the new results, analysis, and clarifications into the revised paper.

**Additional Comments On Reviewer Discussion:**

During the discussion, reviewers raised three main concerns. First, they questioned the use of an image-trained VAE instead of an audio-specific one. The authors addressed this by implementing an audio VAE, which improved FAD and perceptual quality. Second, replacing Griffin-Lim with a neural vocoder (HiFi-GAN) significantly enhanced audio fidelity and FAD scores. Third, reviewers requested clarification on how compressing video features into a single vector retained temporal information. The authors responded with ablations, including attention pooling, demonstrating the effectiveness of their reducer. These revisions effectively addressed the reviewers' concerns, leading Reviewers GqkQ and mkFe to increase their scores. Given this consensus and the overall positive ratings, I see no justification for overturning the reviewers' judgments and recommend accepting the paper.

---

### Decision · Program_Chairs · 2025-01-22

Accept (Poster)